# Directional convergence and alignment in deep learning

**Ziwei Ji**     **Matus Telgarsky**
{ziweiji2,mjt}@illinois.edu
University of Illinois, Urbana-Champaign

## Abstract

In this paper, we show that although the minimizers of cross-entropy and related classification losses are off at infinity, network weights learned by gradient flow converge *in direction*, with an immediate corollary that network predictions, training errors, and the margin distribution also converge. This proof holds for deep homogeneous networks — a broad class of networks allowing for ReLU, max-pooling, linear, and convolutional layers — and we additionally provide empirical support not just close to the theory (e.g., the AlexNet), but also on non-homogeneous networks (e.g., the DenseNet). If the network further has locally Lipschitz gradients, we show that these gradients also converge in direction, and asymptotically *align* with the gradient flow path, with consequences on margin maximization, convergence of saliency maps, and a few other settings. Our analysis complements and is distinct from the well-known neural tangent and mean-field theories, and in particular makes no requirements on network width and initialization, instead merely requiring perfect classification accuracy. The proof proceeds by developing a theory of unbounded nonsmooth Kurdyka-Łojasiewicz inequalities for functions definable in an o-minimal structure, and is also applicable outside deep learning.

## 1   Introduction

Recent efforts to rigorously analyze the optimization of deep networks have yielded many exciting developments, for instance the neural tangent [Jacot et al., 2018, Du et al., 2018, Allen-Zhu et al., 2018, Zou et al., 2018] and mean-field perspectives [Mei et al., 2019, Chizat and Bach, 2018]. In these works, it is shown that small training or even testing error are possible for wide networks.

The above theories, with finite width networks, usually require the weights to stay close to initialization in certain norms. By contrast, practitioners run their optimization methods as long as their computational budget allows [Shallue et al., 2018], and if the data can be perfectly classified, the parameters are guaranteed to diverge in norm to infinity [Lyu and Li, 2019]. This raises a worry that the prediction surface can continually change during training; indeed, even on simple data, as in Figure 1, the prediction surface continues to change after perfect classification is achieved, and even with large width is not close to the maximum margin predictor from the neural tangent regime. If the prediction surface never stops changing, then the generalization behavior, adversarial stability, and other crucial properties of the predictor could also be unstable.

In this paper, we resolve this worry by guaranteeing stable convergence behavior of deep networks as training proceeds, despite this growth of weight vectors to infinity. Concretely:

1. **Directional convergence:** the parameters converge *in direction*, which suffices to guarantee convergence of many other relevant quantities, such as the *prediction margins*.
2. **Alignment:** when gradients exist, they converge in direction to the parameters, which implies various margin maximization results and saliency map convergence, to name a few.

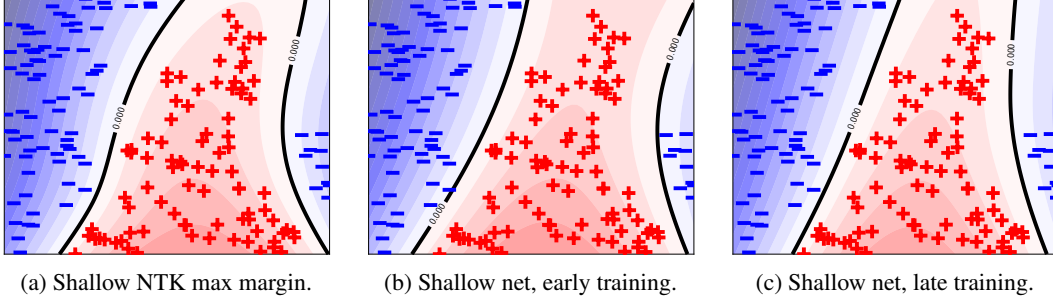

| (a) Shallow NTK max margin. | (b) Shallow net, early training. | (c) Shallow net, late training. |

Figure 1: Prediction surface of a shallow network on simple synthetic data with blue negative examples ("−") and red positive examples ("+"), trained via gradient descent. Figure 1a shows the prediction surface reached by freezing activations, which is also the prediction surface of the corresponding Neural Tangent Kernel (NTK) maximum margin predictor [Soudry et al., 2017]. Figure 1b shows the same network, but now without frozen activations, at the first moment with perfect classification. Training this network much longer converges to Figure 1c.

## 1.1 First result: directional convergence

We show that the network parameters $W_t$ converge *in direction*, meaning the normalized iterates $W_t/\|W_t\|$ converge. Details are deferred to Section 3, but here is a brief overview.

Our networks are *L-positively homogeneous in the parameters*, meaning scaling the parameters by $c > 0$ scales the predictions by $c^L$, and *definable in some o-minimal structure*, a mild technical assumption which we will describe momentarily. Our networks can be arbitrarily deep with many common types of layers (e.g., linear, convolution, ReLU, and max-pooling layers), but homogeneity rules out some components such as skip connections and biases, which all satisfy definability.

We consider binary classification with either the logistic loss $\ell_{\log}(z) := \ln(1 + e^{-z})$ (binary cross-entropy) or the exponential loss $\ell_{\exp}(z) := e^{-z}$, and a standard gradient flow (infinitesimal gradient descent) for non-differentiable non-convex functions via the Clarke subdifferential. We start from an initial risk smaller than $1/n$, where $n$ denotes the number of data samples; in this way, our analysis handles the late phase of training, and can be applied after some other analysis guarantees risk $1/n$.

Under these conditions, we prove the following result, without any other assumptions about the distribution of the parameters or the width of the network (cf. Theorem 3.1):

*The curve swept by $W_t/\|W_t\|$ has finite length, and thus $W_t/\|W_t\|$ converges.*

Our main corollary is that *prediction margins* converge (cf. Corollary 3.2), meaning convergence of the normalized per-example values $y_i\Phi(x_i;W_t)/\|W_t\|^L$, where $y_i$ is the label and $\Phi(x_i;W_t)$ is the prediction on example $x_i$. These quantities are central in the study of generalization of deep networks, and their stability also implies stability of many other useful quantities [Bartlett et al., 2017, Jiang et al., 2019, 2020]. As an illustration of directional convergence and margin convergence, we plot the margin values for all examples in the standard `cifar` data against training iterations in Figure 2; these trajectories exhibit strong convergence behavior, both within our theory (a modified homogeneous AlexNet, as in Figure 2a), and outside of it (DenseNet, as in Figure 2b).

Directional convergence is often assumed throughout the literature [Gunasekar et al., 2018a, Chizat and Bach, 2020], but has only been established for linear predictors [Soudry et al., 2017]. It is tricky to prove because it may still be false for highly smooth functions: for instance, the homogeneous Mexican Hat function satisfies all our assumptions *except* definability, and can be adjusted to have arbitrary order of continuous derivatives, but its gradient flow *does not* converge in direction, instead it spirals [Lyu and Li, 2019]. To deal with similar pathologies in many branches of mathematics, the notion of functions *definable in some o-minimal structure* was developed: these are rich classes of functions built up to limit oscillations and other bad behavior. Using techniques from this literature, we build general tools, in particular unbounded nonsmooth Kurdyka-Łojasiewicz inequalities, which allows us to prove directional convergence, and may also be useful outside deep learning. More discussion on the o-minimal literature is given in Section 1.3, technical preliminaries are introduced in Section 2, and a proof overview is given in Section 3, with full details in the appendices.

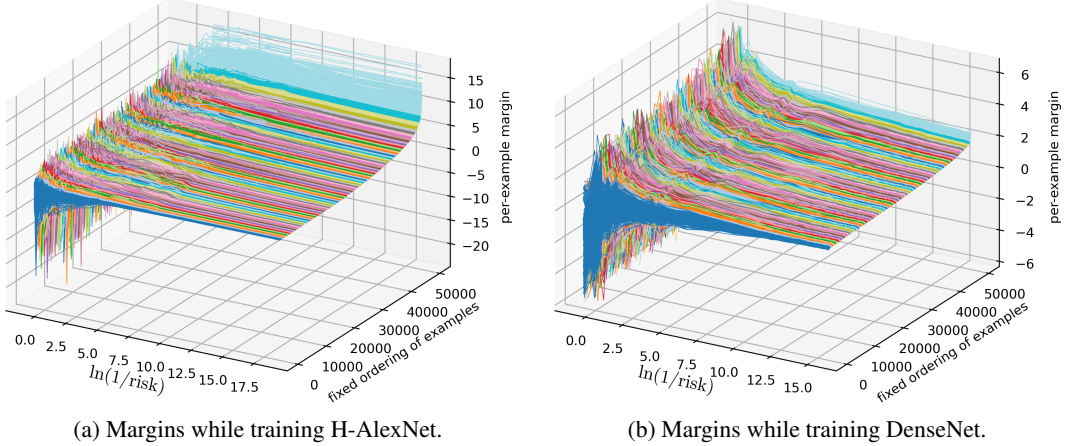

(a) Margins while training H-AlexNet.
(b) Margins while training DenseNet.

Figure 2: The margins of all examples in `cifar`, plotted against time, or rather optimization accuracy $\ln(n/\mathcal{L}(W_t))$ to remove the effect of step size and other implementation coincidences. Figure 2a shows "H-AlexNet", a homogeneous version of AlexNet as described in the main text [Krizhevsky et al., 2012], which is handled by our theory. Figure 2b shows a standard DenseNet [Huang et al., 2017], which does not fit the theory in this work due to skip connections and biases, but still exhibits convergence of margins, thus suggesting a tantalizing open problem.

## 1.2 Second result: gradient alignment

Our second contribution, in Section 4, is that if the network has locally Lipschitz gradients, then these gradients also converge, and are *aligned* to the gradient flow path (cf. Theorem 4.1).

*The gradient flow path, and the gradient of the risk along the path, converge to the same direction.*

As a practical consequence of this, recall the use of gradients within the interpretability literature, specifically in *saliency maps* [Adebayo et al., 2018]: if gradients do not converge in direction then saliency maps can change regardless of the number of iterations used to produce them. As a theoretical consequence, directional convergence and alignment imply margin maximization in a variety of situations: this holds in the deep linear case, strengthening prior work [Gunasekar et al., 2018b, Ji and Telgarsky, 2018a], and in the 2-homogeneous network case, with an assumption taken from the infinite width setting [Chizat and Bach, 2020], but presented here with finite width.

## 1.3 Further related work

Our analysis is heavily inspired and influenced by the work of Lyu and Li [2019], who studied margin maximization of homogeneous networks, establishing monotonicity of a *smoothed margin*, a quantity we also use. However, they did not prove directional convergence but instead must use subsequences. Their work also left open alignment and global margin maximization.

**Directional convergence.** A standard approach to resolve directional convergence and similar questions is to establish that the objective function in question is *definable in some o-minimal structure*, which as mentioned before, limits oscillations and other complicated behavior. This literature cannot be directly applied to our setting, owing to a combination of nonsmooth layers like the ReLU and max-pooling, and the exponential function used in the cross entropy loss, and as a result, our proofs need to rebuild many o-minimal results from the ground up.

In more detail, an important problem in the o-minimal literature is the *gradient conjecture* of René Thom: it asks when the existence of $\lim_{t\to\infty} W_t = z$ further implies $\lim_{t\to\infty} (W_t-z)/\|W_t-z\|$ exists, and was established in various definable scenarios by Kurdyka et al. [2000a, 2006] via related Kurdyka-Łojasiewicz inequalities [Kurdyka, 1998]. The underlying proof ideas can also be used to analyze $\lim_{t\to\infty} W_t/\|W_t\|$ when the weights go to infinity [Grandjean, 2007]. However, the prior results require the objective function to be either real analytic, or definable in a "polynomially-bounded" o-minimal structure. The first case causes the aforementioned nonsmoothness issue, and

excludes many common layers in deep learning such as the ReLU and max-pooling. The second case excludes the exponential function, and means the logistic and cross-entropy losses cannot be handled. To resolve these issues, we had to redo large portions of the o-minimality theory, such as the nonsmooth unbounded Kurdyka-Łojasiewicz inequalities that can handle the exponential/logistic loss, as presented in Section 3.

**Alignment.** As discussed in Section 4, alignment implies the gradient flow reaches a stationary point of the limiting margin maximization objective, and therefore is related to various statements and results throughout the literature on implicit bias and margin maximization [Soudry et al., 2017, Ji and Telgarsky, 2018b]. This stationary point perspective also appears in some nonlinear works, for instance in the aforementioned work on margins by Lyu and Li [2019], which showed that *subsequences* of the gradient flow converge to such stationary points; in addition to fully handling the gradient flow, the present work also differs in that alignment is in general a stronger notion, in that it is unclear how to prove alignment as a consequence of convergence to KKT points. Additionally, alignment can still hold when the objective function is not definable and directional convergence is false, for example on the homogeneous Mexican hat function, which cannot be handled by the approach in [Lyu and Li, 2019, Appendix J]. As a final pointer to the literature, many implicit bias works explicitly assume directional convergence and some version of alignment [Gunasekar et al., 2018b, Chizat and Bach, 2020], but neither do these works indicate a possible proof, nor do they provide conclusive evidence.

### 1.4 Experimental overview

The experiments in Figures 1 and 2 are performed in as standard a way as possible to highlight that directional convergence is a reliable property; full details are in Appendix A. Briefly, Figure 1 uses synthetic data and vanilla gradient descent (no momentum, no weight decay, etc.) on a 10,000 node wide 2-layer *squared* ReLU network and its Neural Tangent Kernel classifier; by using the squared ReLU, both our directional convergence and our alignment results apply. Figure 2 uses standard `cifar` firstly with a modified homogeneous AlexNet and secondly with an unmodified DenseNet, respectively inside and outside our assumptions. SGD was used on `cifar` due to training set size, and seeing how directional convergence still seems to occur, suggests another open problem.

## 2 Preliminaries and assumptions

In this section, we first introduce the notions of Clarke subdifferentials and o-minimal structures, and then use these notions to describe the network model, gradient flow, and Assumptions 2.1 and 2.2. Throughout this paper, $\|\cdot\|$ denotes the $\ell_2$ (Frobenius) norm, and $\|\cdot\|_\sigma$ denotes the spectral norm.

**Locally Lipschitz functions and Clarke subdifferentials.** Consider a function $f : D \to \mathbb{R}$ with $D$ open. We say that $f$ is *locally Lipschitz* if for any $x \in D$, there exists a neighborhood $U$ of $x$ such that $f|_U$ is Lipschitz continuous. We say that $f$ is $C^1$ if $f$ is continuously differentiable on $D$.

If $f$ is locally Lipschitz, it holds that $f$ is differentiable a.e. [Borwein and Lewis, 2000, Theorem 9.1.2]. The *Clarke subdifferential* of $f$ at $x \in D$ is defined as

$$\partial f(x) := \text{conv} \left\{ \lim_{i \to \infty} \nabla f(x_i) \middle| x_i \in D, \nabla f(x_i) \text{ exists}, \lim_{i \to \infty} x_i = x \right\},$$

which is nonempty convex compact [Clarke, 1975], and if $f$ is continuously differentiable at $x$, then $\partial f(x) = \{\nabla f(x)\}$. Vectors in $\partial f(x)$ are called *subgradients*, and we let $\bar{\partial} f(x)$ denote the unique minimum-norm subgradient:

$$\bar{\partial} f(x) := \underset{x^* \in \partial f(x)}{\arg\min} \|x^*\|.$$

In the following analysis, we use $\bar{\partial} f$ in many places that seem to call on $\nabla f$.

**O-minimal structures and definable functions.** Formally, an o-minimal structure is a collection $\mathcal{S} = \{\mathcal{S}_n\}_{n=1}^\infty$, where $\mathcal{S}_n$ is a set of subsets of $\mathbb{R}^n$ which includes all algebraic sets and is closed under finite union/intersection and complement, Cartesian product, and projection, and $\mathcal{S}_1$ consists

of finite unions of open intervals and points. A set $A \subset \mathbb{R}^n$ is *definable* if $A \in \mathcal{S}_n$, and a function $f : D \to \mathbb{R}^m$ with $D \subset \mathbb{R}^n$ is *definable* if its graph is in $\mathcal{S}_{n+m}$. More details are given in Appendix B.

Many natural functions and operations are definable. First of all, definability of functions is stable under algebraic operations, composition, inverse, maximum and minimum, etc. Moreover, Wilkie [1996] proved that there exists an o-minimal structure where polynomials and the exponential function are definable. Consequently, definability allows many common layer types in deep learning, such as fully-connected/convolutional/ReLU/max-pooling layers, skip connections, the cross entropy loss, etc.; moreover, they can be composed arbitrarily As will be discussed later, what is still missing is the handling of the gradient flow on such functions.

**The network model.** Consider a dataset $\{(x_i, y_i)\}_{i=1}^n$, where $x_i \in \mathbb{R}^d$ are features and $y_i \in \{-1, +1\}$ are binary labels, and a predictor $\Phi(\cdot; W) : \mathbb{R}^d \to \mathbb{R}$ with parameters $W \in \mathbb{R}^k$. We make the following assumption on the predictor $\Phi$.

**Assumption 2.1.** For any fixed $x$, the prediction $W \mapsto \Phi(x; W)$ as a function of $W$ is locally Lipschitz, $L$-positively homogeneous for some $L > 0$, and definable in some o-minimal structure including the exponential function.

As mentioned before, homogeneity means that $\Phi(x; cW) = c^L \Phi(x; W)$ for any $c \geq 0$. This means, for instance, that linear, convolutional, ReLU, and max-pooling layers are permitted, but not skip connections and biases. Homogeneity is used heavily throughout the theoretical study of deep networks [Lyu and Li, 2019].

Given a decreasing loss function $\ell$, the total loss (or *unnormalized empirical risk*) is given by

$$\mathcal{L}(W) := \sum_{i=1}^n \ell \left( y_i \Phi(x_i; W) \right) = \sum_{i=1}^n \ell(p_i(W)),$$

where $p_i(W) := y_i \Phi(x_i; W)$ are also locally Lipschitz, $L$-positively homogeneous and definable under Assumption 2.1. We consider the exponential loss $\ell_{\exp}(z) := e^{-z}$ and the logistic loss $\ell_{\log}(z) := \ln(1 + e^{-z})$, in which case $\mathcal{L}$ is also locally Lipschitz and definable.

**Gradient flow.** As in [Davis et al., 2020, Lyu and Li, 2019], a curve $z$ from an interval $I$ to some real space $\mathbb{R}^m$ is called an *arc* if it is absolutely continuous on any compact subinterval of $I$. It holds that an arc is a.e. differentiable, and the composition of an arc and a locally Lipschitz function is still an arc. We consider a gradient flow $W : [0, \infty) \to \mathbb{R}^k$ that is an arc and satisfies

$$\frac{\mathrm{d}W_t}{\mathrm{d}t} \in -\partial \mathcal{L}(W_t), \quad \text{for a.e. } t \geq 0. \tag{1}$$

Our second assumption is on the initial risk, and appears in prior work [Lyu and Li, 2019].

**Assumption 2.2.** The initial iterate $W_0$ satisfies $\mathcal{L}(W_0) < \ell(0)$.

As mentioned before, this assumption encapsulates our focus on the "late training" phase; some other analysis, for instance the neural tangent kernel, can be first applied to ensure $\mathcal{L}(W_0) < \ell(0)$.

## 3  Directional convergence

We now turn to stating our main result on directional convergence and sketching its analysis. As Assumptions 2.1 and 2.2 imply $\|W_t\| \to \infty$ [Lyu and Li, 2019], we study the normalized flow $\widetilde{W}_t := W_t / \|W_t\|$, whose convergence is a formal way of studying the directional convergence of $W_t$. As mentioned before, directional convergence is false in general [Lyu and Li, 2019], but definability suffices to ensure it. Throughout, for general nonzero $W$, we will use $\widetilde{W} := W / \|W\|$.

**Theorem 3.1.** *Under Assumptions 2.1 and 2.2, for $\ell_{\exp}$ and $\ell_{\log}$, the curve swept by $\widetilde{W}_t$ has finite length, and thus $\widetilde{W}_t$ converges.*

A direct consequence of Theorem 3.1 is the convergence of the *margin distribution* (i.e., normalized outputs). Due to homogeneity, for any nonzero $W$, we have $p_i(W) / \|W\|^L = p_i(\widetilde{W})$, and thus the next result follows from Theorem 3.1.

**Corollary 3.2.** *Under Assumptions 2.1 and 2.2, for $\ell_{\exp}$ and $\ell_{\log}$, it holds that $p_i(W_t)/\|W_t\|^L$ converges for all $1 \le i \le n$.*

Next we give a proof sketch of Theorem 3.1; the full proofs of the Kurdyka-Łojasiewicz inequalities (Lemmas 3.5 and 3.6) are given in Appendix B.3, while the other proofs are given in Appendix C.

### 3.1 A proof sketch of Theorem 3.1

The *smoothed margin* introduced in [Lyu and Li, 2019] is crucial in our analysis: given $W \ne 0$, let

$$\alpha(W) := \ell^{-1}\left(\mathcal{L}(W)\right), \quad \text{and} \quad \tilde{\alpha}(W) := \frac{\alpha(W)}{\|W\|^L}.$$

For simplicity, let $\tilde{\alpha}_t$ denote $\tilde{\alpha}(W_t)$, and $\zeta_t$ denote the length of the path swept by $\widetilde{W}_t = W_t/\|W_t\|$ from time 0 to $t$. Lyu and Li [2019] proved that $\tilde{\alpha}_t$ is nondecreasing with some limit $a \in (0, \infty)$, and $\|W_t\| \to \infty$. We invoke a standard but sophisticated tool from the definability literature to aid in proving $\zeta_t$ is finite: formally, a function $\Psi : [0, \nu) \to \mathbb{R}$ is called a *desingularizing function* when $\Psi$ is continuous on $[0, \nu)$ with $\Psi(0) = 0$, and continuously differentiable on $(0, \nu)$ with $\Psi' > 0$; in words, a desingularizing function is a *witness* to the fact that the flow is asymptotically well-behaved. As we will sketch after stating the lemma, this immediately leads to a proof of Theorem 3.1.

**Lemma 3.3.** *There exist $R > 0$, $\nu > 0$ and a definable desingularizing function $\Psi$ on $[0, \nu)$, such that for a.e. large enough $t$ with $\|W_t\| > R$ and $\tilde{\alpha}_t > a - \nu$, it holds that*

$$\frac{\mathrm{d}\zeta_t}{\mathrm{d}t} \le -c\frac{\mathrm{d}\Psi\left(a - \tilde{\alpha}_t\right)}{\mathrm{d}t}$$

*for some constant $c > 0$.*

To prove Theorem 3.1 from here, let $t_0$ be large enough so that the conditions of Lemma 3.3 hold for all $t \ge t_0$: then we have $\lim_{t\to\infty} \zeta_t \le \zeta_{t_0} + c\Psi\left(a - \tilde{\alpha}_{t_0}\right) < \infty$, and thus the path length is finite.

Below we sketch the proof of Lemma 3.3, which is based on a careful comparison of $\mathrm{d}\tilde{\alpha}_t/\mathrm{d}t$ and $\mathrm{d}\zeta_t/\mathrm{d}t$. The proof might be hard to parse due to the extensive use of $\bar{\partial}$, the minimum-norm Clarke subgradient; at first reading, the condition of local Lipschitz continuity can just be replaced with continuous differentiability, in which case the Clarke subgradient is just the normal gradient.

Given any function $f$ which is locally Lipschitz around a nonzero $W$, let

$$\bar{\partial}_r f(W) := \left\langle \bar{\partial} f(W), \widetilde{W} \right\rangle \widetilde{W} \quad \text{and} \quad \bar{\partial}_\perp f(W) := \bar{\partial} f(W) - \bar{\partial}_r f(W)$$

denote the radial and spherical parts of $\bar{\partial} f(W)$ respectively. First note the following technical characterization of $\mathrm{d}\tilde{\alpha}_t/\mathrm{d}t$ and $\mathrm{d}\zeta_t/\mathrm{d}t$ using the radial and spherical components of relevant Clarke subgradients.

**Lemma 3.4.** *It holds for a.e. $t \ge 0$ that*

$$\frac{\mathrm{d}\tilde{\alpha}_t}{\mathrm{d}t} = \left\|\bar{\partial}_r \tilde{\alpha}(W_t)\right\|\left\|\bar{\partial}_r \mathcal{L}(W_t)\right\| + \left\|\bar{\partial}_\perp \tilde{\alpha}(W_t)\right\|\left\|\bar{\partial}_\perp \mathcal{L}(W_t)\right\|, \quad \text{and} \quad \frac{\mathrm{d}\zeta_t}{\mathrm{d}t} = \frac{\left\|\bar{\partial}_\perp \mathcal{L}(W_t)\right\|}{\|W_t\|}.$$

For simplicity, in the discussion here we consider the case that all subgradients in Lemma 3.4 are nonzero, with the general case handled in the full proofs in the appendices. Then Lemma 3.4 implies

$$\frac{\mathrm{d}\tilde{\alpha}_t}{\mathrm{d}\zeta_t} = \frac{\mathrm{d}\tilde{\alpha}_t/\mathrm{d}t}{\mathrm{d}\zeta_t/\mathrm{d}t} = \|W_t\|\left(\frac{\left\|\bar{\partial}_r \mathcal{L}(W_t)\right\|}{\left\|\bar{\partial}_\perp \mathcal{L}(W_t)\right\|}\left\|\bar{\partial}_r \tilde{\alpha}(W_t)\right\| + \left\|\bar{\partial}_\perp \tilde{\alpha}(W_t)\right\|\right). \tag{2}$$

As in [Kurdyka et al., 2006, Grandjean, 2007], to bound eq. (2), we further consider two cases depending on the ratio $\left\|\bar{\partial}_\perp \tilde{\alpha}(W_t)\right\| / \left\|\bar{\partial}_r \tilde{\alpha}(W_t)\right\|$.

If $\left\|\bar{\partial}_\perp \tilde{\alpha}(W_t)\right\| / \left\|\bar{\partial}_r \tilde{\alpha}(W_t)\right\| \ge c_1\|W_t\|^{L/3}$ for some constant $c_1 > 0$, then Lemma 3.3 follows from $\mathrm{d}\tilde{\alpha}_t/\mathrm{d}\zeta_t \ge \|W_t\|\left\|\bar{\partial}_\perp \tilde{\alpha}(W_t)\right\|$ as given by eq. (2), and the following Kurdyka-Łojasiewicz inequality. Its proof is based on the proof idea of [Kurdyka et al., 2006, Proposition 6.3], but further handles the unbounded and nonsmooth setting.

**Lemma 3.5.** *Given a locally Lipschitz definable function $f$ with an open domain $D \subset \{x \mid \|x\| > 1\}$, for any $c, \eta > 0$, there exists $\nu > 0$ and a definable desingularizing function $\Psi$ on $[0, \nu)$ such that*

$$\Psi'\left(f(x)\right)\|x\|\|\bar{\partial}f(x)\| \geq 1, \quad \text{if } f(x) \in (0, \nu) \text{ and } \|\bar{\partial}_\perp f(x)\| \geq c\|x\|^\eta \|\bar{\partial}_r f(x)\|.$$

On the other hand, if $\|\bar{\partial}_\perp \tilde{\alpha}(W_t)\| / \|\bar{\partial}_r \tilde{\alpha}(W_t)\| \leq c_1 \|W_t\|^{L/3}$, then a careful calculation (using Lemmas C.2 to C.4) can show that for some constants $c_2, c_3 > 0$,

$$\frac{\|\bar{\partial}_r \mathcal{L}(W_t)\|}{\|\bar{\partial}_\perp \mathcal{L}(W_t)\|} \geq c_2 \|W_t\|^{2L/3}, \quad \text{and} \quad \frac{\|\bar{\partial}_r \tilde{\alpha}(W_t)\|}{\|\bar{\partial}\tilde{\alpha}(W_t)\|} \geq c_3 \|W_t\|^{-L/3}.$$

It then follows from eq. (2) that $\mathrm{d}\tilde{\alpha}_t / \mathrm{d}\zeta_t \geq c_2 c_3 \|W_t\|^{4L/3} \|\bar{\partial}\tilde{\alpha}(W_t)\|$. In this case we give the following Kurdyka-Łojasiewicz inequality, which implies Lemma 3.3.

**Lemma 3.6.** *Given a locally Lipschitz definable function $f$ with an open domain $D \subset \{x \mid \|x\| > 1\}$, for any $\lambda > 0$, there exists $\nu > 0$ and a definable desingularizing function $\Psi$ on $[0, \nu)$ such that*

$$\max\left\{1, \frac{2}{\lambda}\right\} \Psi'\left(f(x)\right)\|x\|^{1+\lambda} \|\bar{\partial}f(x)\| \geq 1, \quad \text{if } f(x) \in (0, \nu).$$

# 4 Alignment between the gradient flow path and gradients

Theorem 3.1 gave our directional convergence result, namely that the normalized iterate $W_t / \|W_t\|$ converges to some direction. Next we show and discuss our alignment result, that if all $p_i$ have locally Lipschitz gradients, then along the gradient flow path, $-\nabla \mathcal{L}(W_t)$ converges to the same direction as $W_t$.

**Theorem 4.1.** *Under Assumptions 2.1 and 2.2, if all $p_i$ further have locally Lipschitz gradients, then $-\nabla \mathcal{L}(W_t)$ and $W_t$ converge to the same direction, meaning the angle between $W_t$ and $-\nabla \mathcal{L}(W_t)$ converges to zero. If all $p_i$ are twice continuously differentiable, then the same result holds without the definability condition (cf. Assumption 2.1).*

Below we first sketch the proof of Theorem 4.1, with full details in Appendix D, and then in Section 4.2 present a few global margin maximization consequences, which are proved in Appendix E.

## 4.1 A proof sketch of Theorem 4.1

Recall that $\lim_{t \to \infty} \alpha(W_t) / \|W_t\|^L = a$. The first observation is that $\alpha(W_t)$, the smoothed margin function, asymptotes to the exact margin $\min_{1 \leq i \leq n} p_i(W_t)$ which is $L$-positively homogeneous. Therefore $\alpha$ is asymptotically $L$-positively homogeneous, and formally we can show

$$\lim_{t \to \infty} \left\langle \frac{\nabla \alpha(W_t)}{\|W_t\|^{L-1}}, \frac{W_t}{\|W_t\|} \right\rangle = \lim_{t \to \infty} \frac{\langle \nabla \alpha(W_t), W_t \rangle}{\|W_t\|^L} = aL, \tag{3}$$

which can be viewed as an asymptotic version of Euler's homogeneous function theorem (cf. Lemma C.1). Consequently, the inner product between $\nabla \alpha(W_t) / \|W_t\|^{L-1}$ and $\widetilde{W}_t$ converges.

Let $\theta_t$ denote the angle between $W_t$ and $-\nabla \mathcal{L}(W_t)$, which is also the angle between $W_t$ and $\nabla \alpha(W_t)$, since $\nabla \mathcal{L}(W_t)$ and $\nabla \alpha(W_t)$ point to opposite directions by the chain rule. By [Lyu and Li, 2019, Corollary C.10], given any $\epsilon > 0$, there exists a time $t_\epsilon$ such that $\theta_{t_\epsilon} < \epsilon$. The question is whether such a small angle can be maintained after $t_\epsilon$. This is not obvious since, as mentioned above, the smoothed margin $\alpha(W_t)$ asymptotes to the exact margin $\min_{1 \leq i \leq n} p_i(W_t)$, which may be nondifferentiable even with smooth $p_i$, due to nondifferentiability of the minimum. Consequently, the exact margin may have discontinuous Clarke subdifferentials, and since the smoothed margin asymptotes to it, it is unclear whether $\theta_t \to 0$. (This point was foreshadowed earlier, where it was pointed out that alignment is not a clear consequence of convergence to stationary points of the margin maximization objective.)

To handle this, the key to our analysis is the potential function $\mathcal{J}(W) := \|\nabla \alpha(W_t)\|^2 / \|W_t\|^{2L-2}$. Suppose at time $t$, it holds that $\left\langle \nabla \alpha(W_t) / \|W_t\|^{L-1}, \widetilde{W}_t \right\rangle$ is close to $aL$, and $\theta_t$ is very small. If $\theta_{t'}$

becomes large again at some $t' > t$, it must follows that $\mathcal{J}(W_{t'})$ is much larger than $\mathcal{J}(W_t)$. We prove that this is impossible, by showing that

$$\lim_{t \to \infty} \int_t^\infty \frac{\mathrm{d}\mathcal{J}(W_\tau)}{\mathrm{d}\tau} \, \mathrm{d}\tau = 0, \tag{4}$$

and thus Theorem 4.1 follows. The proof of eq. (4) is motivated by the dual convergence analysis in [Ji and Telgarsky, 2019], and also uses the positive homogeneity of $\nabla p_i$ and $\nabla^2 p_i$ (which exist a.e.).

## 4.2 Main alignment consequence: margin maximization

A variety of (global) margin maximization results are immediate consequences of directional convergence and alignment. This subsection investigates two examples: deep linear networks, and shallow squared ReLU networks.

Deep linear networks predict with $\Phi(x_i; W) = A_L \cdots A_1 x_i$, where the parameters $W = (A_L, \ldots, A_1)$ are organized into $L$ matrices. This setting has been considered in the literature, but the original work assumed directional convergence, alignment and a condition on the support vectors [Gunasekar et al., 2018b]; a follow-up dropped the directional convergence and alignment assumptions, but instead assumed the support vectors span the space $\mathbb{R}^d$ [Ji and Telgarsky, 2018a]. As follows, we not only drop the all aforementioned assumptions, but moreover include a *proof* rather than an assumption of directional convergence.

**Proposition 4.2.** *Suppose $W_t = (A_L(t), \ldots, A_1(t))$ and $\mathcal{L}(W_0) < \ell(0)$. Then a unique linear max margin predictor $\bar{u} := \arg\max_{\|u\| \leq 1} \min_i y_i x_i^\intercal u$ exists, and there exist unit vectors $(v_L, \ldots, v_1, v_0)$ with $v_L = 1$ and $v_0 = \bar{u}$ such that*

$$\lim_{t \to \infty} \frac{A_j(t)}{\|A_j(t)\|} = v_j v_{j-1}^\intercal \qquad \textit{and} \qquad \lim_{t \to \infty} \frac{A_L(t) \cdots A_1(t)}{\|A_L(t) \cdots A_1(t)\|} = \bar{u}^\intercal.$$

Thanks to directional convergence and alignment (cf. Theorems 3.1 and 4.1), the proof boils down to writing down the gradient expression for each layer and doing some algebra.

A more interesting example is a certain 2-homogeneous case, which despite its simplicity is a universal approximator; this setting was studied by Chizat and Bach [2020], who considered the infinite width case, and established margin maximization under *assumptions* of directional convergence and gradient convergence. Unfortunately, it is not clear if Theorems 3.1 and 4.1 can be applied to fill these assumptions, since they do not handle infinite width, and indeed it is not clear if infinite width networks or close relatives are definable in an o-minimal structure. Instead, here we consider the finite width case, albeit with an additional assumption.

Following [Chizat and Bach, 2020, S-ReLU], organize $W_t$ into $m$ rows $(w_j(t))_{j=1}^m$, with normalizations $\theta_j(t) := w_j(t)/\|w_j(t)\|$ where $\theta_j(t) = 0$ when $\|w_j(t)\| = 0$, and consider

$$\Phi(x_i; W) := \sum_j (-1)^j \max\{0, w_j^\intercal x_i\}^2 \quad \text{and} \quad \varphi_{ij}(w) := y_i(-1)^j \max\{0, w^\intercal x_i\}^2, \tag{5}$$

whereby $p_i(W) = \sum_j \varphi_{ij}(w_j)$, and $\Phi$, $p_i$, and $\varphi_{ij}$ are all 2-homogeneous and definable. (The "$(-1)^j$" may seem odd, but is an easy trick to get universal approximation without outer weights.)

**Proposition 4.3.** *Consider the setting in eq. (5) along with $\mathcal{L}(W_0) < \ell(0)$ and $\|x_i\| \leq 1$.*

1. *(**Local guarantee.**) $s \in \mathbb{R}^m$ with $s_j(t) := \|w_j(t)\|^2/\|W_t\|^2$ satisfies $s \to \bar{p} \in \Delta_m$ (probability simplex on $m$ vertices), and $\theta_j \to \bar{\theta}_j$ with $\bar{\theta}_j = 0$ if $s_j = 0$, and*

$$a = \lim_{t \to \infty} \min_i \frac{p_i(W_t)}{\|W_t\|^2} = \lim_{t \to \infty} \min_i \sum_j s_j(t)\varphi_{ij}(\theta_j(t)) = \min_i \max_{s \in \Delta_m} \sum_j s_j \varphi_{ij}(\bar{\theta}_j).$$

2. *(**Global guarantee.**) Suppose the* covering condition*: there exist $t_0$ and $\epsilon > 0$ with*

$$\max_j \|\theta_j(t_0) - \bar{\theta}_j\|_2 \leq \epsilon, \ \textit{and} \ \max_{\theta' \in \mathbb{S}^{d-1}} \max\left\{ \min_{2|j} \|\theta_j(t_0) - \theta'\|, \min_{2 \nmid j} \|\theta_j(t_0) - \theta'\| \right\} \leq \epsilon,$$

*where $\mathbb{S}^{d-1} := \{\theta \in \mathbb{R}^d : \|\theta\| = 1\}$. Then margins are approximately (globally) maximized:*

$$\lim_{t \to \infty} \min_i \frac{p_i(W_t)}{\|W_t\|^2} \geq \max_{\nu \in \mathcal{P}(\mathbb{S}^{d-1})} \min_i y_i \int \max\{0, x_i^\intercal \theta\}^2 \, \mathrm{d}\nu(\theta) - 4\epsilon,$$

*where $\mathcal{P}(\mathbb{S}^{d-1})$ is the set of* signed *measures on $\mathbb{S}^{d-1}$ with mass at most 1.*

The first part (the "local guarantee") characterizes the limiting margin as the maximum margin of a *linear* problem obtained by taking the limiting directions $(\bar{\theta}_j)_{j=1}^m$ and treating the resulting $\varphi_{ij}(\bar{\theta}_j)$ as features. The quality of this margin is bad if the limiting directions are bad, and therefore we secondly (the "global guarantee") consider a case where our margin is nearly as good as the *infinite width global max margin value* as defined by [Chizat and Bach, 2020, eq. (5)]; see discussion therein for a justification of this choice, and moreover calling it the globally maximal margin.

The *covering condition* deserves further discussion. In the infinite width setting, it holds for all $\epsilon > 0$ assuming directional convergence [Chizat and Bach, 2020, Proof of Theorem D.1], but cannot hold in such generality here as we are dealing with finite width. Similar properties have appeared throughout the literature: Wei et al. [2018, Section 3] explicitly re-initialized network nodes to guarantee a good covering, and more generally [Ge et al., 2015] added noise to escape saddle points in general optimization problems.

## 5 Concluding remarks and open problems

In this paper, we established that the normalized parameter vectors $W_t/\|W_t\|$ converge, and that under an additional assumption of locally Lipschitz gradients, the gradients also converge and align with the parameters.

There are many promising avenues for future work based on these results. One basic line is to weaken our assumptions: dropping homogeneity to allow for DenseNet and ResNet, and analyzing finite-time methods like (stochastic) gradient descent, and moreover their rates of convergence. We also handled only the binary classification case, however our tools should directly allow for cross-entropy.

Another direction is into further global margin maximization results, beyond the simple networks in Section 4.2, and into related generalization consequences of directional convergence and alignment.

## Broader impact

This paper constitutes theoretical work, with an aim of enhancing human understanding, and laying the groundwork for further theoretical and applied work. The authors hope that advancing the foundations of deep networks leads moreover to a better understanding of their failure modes, and manipulation thereof, and thus an increase in safety.

## Acknowledgments and disclosure of funding

The authors thank Zhiyuan Li and Kaifeng Lyu for lively discussions during an early phase of the project. The authors are grateful for support from the NSF under grant IIS-1750051, and from NVIDIA under a GPU grant.

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
