[Supplementary Material]

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

# A Experimental setup

The goal of the experiments is to illustrate that directional convergence is a clear, reliable phenomenon. Below we detail the setup for the two types of experiments: contour plots in Figure 1, and margin plots in Figure 2 (with ResNet here in Figure 3).

**Data.** Figure 1 used two-dimensional synthetic data in order to capture the entire prediction surface; data was generated by labeling points in the plane with a random network (which included a bias term), and then deleting low-margin points. Then, when training from scratch to produce the contours, data was embedded in $\mathbb{R}^3$ by appending a 1; this added bias made the maximum margin network much simpler.

Figure 2 used the standard `cifar` dataset in its 10 class configuration [Krizhevsky, 2009]. There are 50,000 data points, each with 3072 dimensions, organized into $32 \times 32$ images with 3 color channels.

**Models.** A few simple models both inside and outside our technical assumptions were used. All code was implemented in PyTorch [Paszke et al., 2019].

Figure 1 worked with a style of 2-layer network which appears widely throughout theoretical investigations: specifically, there is first a wide linear layer (in our case, 10,000 nodes), then a *squared* ReLU layer, and then a layer of random signs which is not trained. This squared ReLU network with one trainable layer is 2-homogeneous, and was chosen both to fit with the alignment guarantee in Theorem 4.1, and also to amplify differences with the NTK. Note that this simple architecture is still a universal approximator with non-convex training. Figures 1b and 1c trained this network, which can be written as $x \mapsto \sum_j s_j \max\left\{0, \langle w_j, x \rangle\right\}^2$, where $s_j \in \pm 1$ are fixed random signs and $(w_j)_{j=1}^m$ are the trainable parameters. Figure 1a trained the corresponding NTK [Jacot et al., 2018, Du et al., 2018, Allen-Zhu et al., 2018, Zou et al., 2018], meaning the linear predictor obtained by freezing the network activations, which thus has the form $x \mapsto \sum_j s_j \langle v_j, x \rangle \max\{0, \langle w_j, x \rangle\}$, where $(w_j)_{j=1}^m$ from before are now fixed, and only $(v_j)_{j=1}^m$ are trained.

Figure 2 used convolutional networks. Firstly, Figure 2a used "H-AlexNet", which is based on a simplified version of the standard AlexNet [Krizhevsky et al., 2012] as presented in the PyTorch `cifar` tutorial [Paszke et al., 2019], but with biases disabled in order to give a homogeneous network. The network ultimately consists of ReLU layers, max-pooling layers, linear layers, and convolutional layers, and is 5-homogeneous. In particular, H-AlexNet satisfies all conditions we need for directional convergence.

The two models outside the assumptions were DenseNet (cf. Figure 2b and ResNet (cf. Figure 3), used unmodified from the PyTorch source, namely by invoking `torchvision.models.densetnet121` and `torchvision.models.resnet18` with argument `num_classes=10`.

**Training.** Training was a basic gradient descent (GD) for Figure 1, and a basic stochastic gradient descent (SGD) for Figures 2 and 3 with a mini-batch size of 512; there was no weight decay or other regularization, no momentum, etc.; it is of course an interesting question how more sophisticated optimization schemes, including AdaGrad and AdaDelta and others, affect directional convergence and alignment. Experiments were run to accuracy $10^{-8}$ or greater in order to train significantly past the point $\mathcal{L}(W_0) < \ell(0)$ from Assumption 2.2, and to better depict directional convergence.

To help reach such small risk, the main ideas were to rewrite the objective functions to be numerically stable, and secondly to scale the step size by $1/\mathcal{L}(W_{t-1})$, which incidentally is consistent with gradient flow on $\alpha$ with exponential loss, and is moreover an idea found across the margin literature, most notably as the step size used in AdaBoost [Freund and Schapire, 1997]. This can lead to some numerical instability, so the step size was reduced if the norm of the induced update was too large, meaning the norm of the gradient times the step size was too large. A much more elaborate numerical scheme was reported by Lyu and Li [2019, Appendix L], but not used here.

One point worth highlighting is the role of SGD, which seems as though it should have introduced a great deal of noise into the plots, and after all is outside the assumptions of the paper (which requires gradient flow, let alone gradient descent). Though not depicted here, experiments in Figure 2 were also tried on subsampled data and full gradients, and Figure 1 was tried with SGD in place of GD;

Figure 3: ResNet margins over time, plotted in the same way as Figure 2; see Appendix A for details.

while gradient descent does result in smoother plots, the difference is small overall, leaving the rigorous analysis of directional convergence with SGD as a promising future direction.

**Margin plots.** A few further words are in order for the margin plots in Figures 2 and 3.

While margins are well-motivated from generalization and other theoretical perspectives [Bartlett et al., 2017, Jiang et al., 2019, 2020], we also use margin plots as a visual surrogate for prediction surface contour plots from Figure 1, but now for high-dimensional data, even with high-dimensional outputs. In particular, Figures 2 and 3 track the prediction surface but restricted to the training set, showing, in a sense, the output trajectory for each data example. Since the output dimension is 10 classes, we convert this to a single real number via the usual multi-class margin $(x, y) \mapsto \Phi(x; W_t)_y - \max_{j \neq y} \Phi(x; W_t)_j$.

In the case of homogeneous networks, it is natural to normalize this quantity by $\|W_t\|^L$; for the inhomogeneous cases DenseNet and ResNet, no such normalization is available. Therefore, for consistency, at each time $t$, margins were normalized by the median nonnegative margin across all data.

To show the evolution of the margins most clearly, we sorted margins according to the final margin level, and used this fixed data ordering for all time; as a result, lines in the plot indeed correspond to trajectories of single examples. Moreover, we indexed time by the log of the inverse risk, namely $\ln n/\mathcal{L}(W_t)$ in our notation. While this may seem odd at first, importantly it washes out the effect of small step-sizes and other implementation choices; and crucially disallows an artificial depiction of directional convergence by choosing rapidly-vanishing step sizes.

## B  Results on o-minimal structures

An o-minimal structure is a collection $\mathcal{S} = \{\mathcal{S}_n\}_{n=1}^{\infty}$, where each $\mathcal{S}_n$ is a set of subsets of $\mathbb{R}^n$ satisfying the following conditions:

1. $\mathcal{S}_1$ is the collection of all finite unions of open intervals and points.

2. $\mathcal{S}_n$ includes the zero sets of all polynomials on $\mathbb{R}^n$: if $p$ is a polynomial on $\mathbb{R}^n$, then $\{x \in \mathbb{R}^n \,|\, p(x) = 0\} \in \mathcal{S}_n$.

3. $\mathcal{S}_n$ is closed under finite union, finite intersection, and complement.

4. $\mathcal{S}$ is closed under Cartesian products: if $A \in \mathcal{S}_m$ and $B \in \mathcal{S}_n$, then $A \times B \in \mathcal{S}_{m+n}$.

5. $\mathcal{S}$ is closed under projection $\Pi_n$ onto the first $n$ coordinates: if $A \in \mathcal{S}_{n+1}$, then $\Pi_n(A) \in \mathcal{S}_n$.

Given an o-minimal structure $\mathcal{S}$, a set $A \subset \mathbb{R}^n$ is definable if $A \in \mathcal{S}_n$, and a function $f : D \to \mathbb{R}^m$ with $D \subset \mathbb{R}^n$ is definable if the graph of $f$ is in $\mathcal{S}_{n+m}$. Due to the stability under projection, the domain of a definable function is definable. In the following we consider an arbitrary fixed o-minimal structure.

## B.1 Basic properties

A convenient way to construct definable sets and functions is to use *first-order formulas*:

- If $A$ is a definable set, then "$x \in A$" is a first-order formula.

- If $\phi$ and $\psi$ are first-order formulas, then $\phi \wedge \psi$, $\phi \vee \psi$, $\neg\phi$ and $\phi \Rightarrow \psi$ are first-order formulas.

- If $\phi(x,y)$ is a first-order formula where $x \in \mathbb{R}^n$ and $y \in \mathbb{R}^m$, and $A \subset \mathbb{R}^n$ is definable, then $\forall x \in A\phi(x,y)$ and $\exists x \in A\phi(x,y)$ are first-order formulas.

Given a first-order formula, the set of free variables which satisfy the formula is definable [Van den Dries and Miller, 1996, Appendix A]. The following basic properties of definable sets and functions can then be shown (see [Van den Dries and Miller, 1996, Coste, 2000, Lê Loi, 2010]).

1. Given any $\alpha, \beta \in \mathbb{R}$ and any definable functions $f, g : D \to \mathbb{R}$, we have $\alpha f + \beta g$ and $fg$ are definable. If $g \neq 0$ on $D$, then $f/g$ is definable. If $f \geq 0$ on $D$, then $f^{1/\ell}$ is definable for any positive integer $\ell$.

2. Given a function $f : D \to \mathbb{R}^m$, let $f_i$ denote the $i$-th coordinate of its output. Then $f$ is definable if and only if all $f_i$ are definable.

3. Any composition of definable functions is definable.

4. Any coordinate permutation of a definable set is definable. Consequently, if the inverse of a definable function exists, it is also definable.

5. The image and pre-image of a definable set by a definable function is definable. Particularly, given any real-valued definable function $f$, all of $f^{-1}(0)$, $f^{-1}\left((-\infty, 0)\right)$ and $f^{-1}\left((0, \infty)\right)$ are definable.

6. Any combination of finitely many definable functions with disjoint domains is definable. For example, the pointwise maximum and minimum of definable functions are definable.

The proofs are standard and omitted. To illustrate the idea, we give a proof of the following standard result on the infimum and supremum operation.

**Lemma B.1.** *Let $A \subset \mathbb{R}^{n+1}$ be definable and $\Pi_n$ denote the projection onto the first $n$ coordinates. Suppose $\inf \left\{y \,\middle|\, (x,y) \in A\right\} > -\infty$ for all $x \in \Pi_n(A)$, then the function from $\Pi_n(A)$ to $\mathbb{R}$ given by*

$$x \mapsto \inf \left\{y \,\middle|\, (x,y) \in A\right\}$$

*is definable. Consequently, we have:*

1. *Let $f : D \to \mathbb{R}$ be definable and bounded below, and $g : D \to \mathbb{R}^m$ be definable. Then $h : g(D) \to \mathbb{R}$ given by $h(y) := \inf_{x \in g^{-1}(y)} f(x)$ is definable.*

2. *Let $f : D_f \to \mathbb{R}$ and $g : D_g \to \mathbb{R}$ be definable and bounded below, then their infimal convolution $h : D_f + D_g \to \mathbb{R}$ given by*

$$h(z) := \inf \left\{f(x) + g(y) \,\middle|\, x \in D_f, y \in D_g, x + y = z\right\}$$

*is definable.*

3. *A function $f : D \to \mathbb{R}$ is definable if and only if its epigraph is definable.*

4. *Given a definable set $A$, the function $d_A(x) := \inf_{y \in A} \|x - y\|$ is definable, which implies the closure, interior and boundary of $A$ are definable.*

5. *The lower-semicontinuous envelope of a definable function is definable.*

*Proof.* Note that the set

$$A_\ell := \left\{(x, y) \,\middle|\, x \in \Pi_n(A), \text{ and } \forall(x, y') \in A, y \le y'\right\}$$

is definable, since it is given by the following first-order formula:

$$(x, y): \quad x \in \Pi_n(A) \,\wedge\, \forall(x', y') \in A\left((x = x') \Rightarrow (y \le y')\right).$$

Similarly, the set

$$A_{\ell u} := \left\{(x, y) \,\middle|\, x \in \Pi_n(A), \text{ and } \forall(x, y') \in A_\ell, y \ge y'\right\}$$

is definable, and thus so is $A_\ell \cup A_{\ell u}$, which is the graph of the desired function.

Now we prove the remaining claims.

1. Let $G_f$ denote the graph of $f$, and $G_g$ denote the graph of $g$. We can just apply the main claim to the following definable set:

$$(y, z): \quad y \in g(D) \,\wedge\, \exists(x, y') \in G_g \exists(x', z') \in G_f\left((x = x') \wedge (y = y') \wedge (z = z')\right).$$

2. First, the Minkowski sum of two definable sets $A$ and $B$ is definable:

$$z: \quad \exists x \in A \exists y \in B(x + y = z).$$

Then we can just apply the main claim to the Minkowski sum of the graphs of $f$ and $g$.

3. Let $G_f$ denote the graph of $f$. If $G_f$ is definable, then the epigraph is definable:

$$(x, y): \quad x \in D \,\wedge\, \forall(x', y') \in G_f\left((x = x') \Rightarrow (y \ge y')\right).$$

If the epigraph is definable, then $G_f$ is definable due to the main claim.

4. We can just apply the main claim to the set

$$(x, r): \quad \exists y \in A\left(\|x - y\| = r\right).$$

The closure of $A$ is just $d_A^{-1}(0)$. The interior of $A$ is the complement of $d_{A^c}^{-1}(0)$. The boundary is the difference between the closure and interior.

5. The epigraph of the lower-semicontinuous envelope of $f$ is the closure of the epigraph of $f$.

$\square$

As another example, note that the types of networks under discussion are definable.

**Lemma B.2.** *Suppose there exist $k, d_0, d_1, \ldots, d_L > 0$ and $L$ definable functions $(g_1, \ldots, g_L)$ where $g_j : \mathbb{R}^{d_0} \times \cdots \times \mathbb{R}^{d_{j-1}} \times \mathbb{R}^k \to \mathbb{R}^{d_j}$. Let $h_1(x, W) := g_1(x, W)$, and for $2 \le j \le L$,*

$$h_j(x, W) := g_j\left(x, h_1(x, W), \ldots, h_{j-1}(x, W), W\right),$$

*then all $h_j$ are definable. It suffices if each output coordinate of $g_j$ is the minimum or maximum over some finite set of polynomials, which allows for linear, convolutional, ReLU, max-pooling layers and skip connections.*

*Proof.* The definability of $h_j$ can be proved by induction using the fact that definability is preserved under composition. Next, note that the minimum and maximum of a finite set of polynomials is definable. Lastly, note that each output coordinate of linear and convolutional layers can be written as a polynomial of their input and the parameters; each output coordinate of a ReLU layer is the maximum of two polynomials; each output of a max-pooling layer is a maximum of polynomials. Skip connections are allowed by the definition of $h_j$. $\square$

Below are some useful properties of definable functions.

**Proposition B.3** ([Lê Loi, 2010, Exercise 2.7]). *Given a definable function $f : (a, b) \to \mathbb{R}$ where $-\infty \le a < b \le \infty$, it holds that $\lim_{x \to a^+} f(x)$ and $\lim_{x \to b^-} f(x)$ exist in $\mathbb{R} \cup \{-\infty, +\infty\}$.*

*Proof.* We consider $\lim_{x \to a^+} f(x)$ where $a \in \mathbb{R}$; the other cases can be handled similarly. If $\lim_{x \to a^+} f(x)$ does not exist, then there exists $k \in \mathbb{R}$ such that $\limsup_{x \to a^+} f(x) > k > \liminf_{x \to a^+} f(x)$. In other words, for any $\epsilon > 0$, there exists $x_1, x_2 \in (a, a+\epsilon)$ such that $f(x_1) > k$ and $f(x_2) < k$. However, since $g := f - k$ is definable on $(a, b)$, it holds that $g^{-1}\left((-\infty, 0)\right)$, and $g^{-1}(0)$, and $g^{-1}\left((0, \infty)\right)$ are all definable, and thus they are all finite unions of open intervals and points. It then follows that there exists $\epsilon_0 > 0$ such that $g = f - k$ has a constant sign (i.e., $> 0$, $= 0$ or $< 0$) on $(a, a + \epsilon_0)$, which is a contradiction. $\qquad\square$

**Theorem B.4** (Monotonicity Theorem [Van den Dries and Miller, 1996, Theorem 4.1]). *Given a definable function $f : (a, b) \to \mathbb{R}$ where $-\infty \leq a < b \leq \infty$, there exist $a_0, \ldots, a_k, a_{k+1}$ with $a = a_0 < a_1 < \ldots < a_k < a_{k+1} = b$ such that for all $0 \leq i \leq k$, it holds on $(a_i, a_{i+1})$ that $f$ is $C^1$ and $f'$ has a constant sign (i.e., $> 0$, $= 0$ or $< 0$).*

Proposition B.3 and Theorem B.4 imply the following result which we need later.

**Lemma B.5.** *Given a $C^1$ definable curve $\gamma : [0, \infty) \to \mathbb{R}^n$ such that $\lim_{s \to \infty} \gamma(s)$ exists and is finite, it holds that the path swept by $\gamma$ has finite length.*

*Proof.* Let $z := \lim_{s \to \infty} \gamma(s)$. Since $\|z - \gamma(s)\|$ is definable, either it is $0$ for all large enough $s$, or it is positive for all large enough $s$. In the first case, since $\gamma$ is $C^1$, it has finite length. In the second case, Theorem B.4 implies that there exists an interval $[a, \infty)$ on which $\|z - \gamma(s)\| > 0$ and $\mathrm{d}\|z - \gamma(s)\| / \mathrm{d}s < 0$, and thus $\|\gamma'(s)\| > 0$. Let

$$\lim_{s \to \infty} \frac{z - \gamma(s)}{\|z - \gamma(s)\|} = u, \quad \text{and} \quad \lim_{s \to \infty} \frac{\gamma'(s)}{\|\gamma'(s)\|} = v.$$

The existence of the above limits is guaranteed by Proposition B.3. Note that $\langle u, v \rangle$ is equal to

$$\lim_{s \to \infty} \left\langle \frac{z - \gamma(s)}{\|z - \gamma(s)\|}, v \right\rangle = \lim_{s \to \infty} \frac{\int_s^\infty \langle \gamma'(\tau), v \rangle \, \mathrm{d}\tau}{\|z - \gamma(s)\|} = \lim_{s \to \infty} \frac{\int_s^\infty \|\gamma'(\tau)\| \left\langle \gamma'(\tau)/\|\gamma'(\tau)\|, v \right\rangle \mathrm{d}\tau}{\|z - \gamma(s)\|}.$$

Since $\gamma'(s)/\|\gamma'(s)\| \to v$, given any $\epsilon > 0$, for large enough $s$ it holds that $\left\langle \gamma'(s)/\|\gamma'(s)\|, v \right\rangle \geq 1 - \epsilon$, and thus

$$\langle u, v \rangle = \lim_{s \to \infty} \frac{\int_s^\infty \|\gamma'(\tau)\| \left\langle \gamma'(\tau)/\|\gamma'(\tau)\|, v \right\rangle \mathrm{d}\tau}{\|z - \gamma(s)\|} \geq (1 - \epsilon) \lim_{s \to \infty} \frac{\int_s^\infty \|\gamma'(\tau)\| \mathrm{d}\tau}{\|z - \gamma(s)\|} \geq 1 - \epsilon,$$

which implies that $u = v$. Since $\epsilon > 0$ was arbitrary, then

$$\lim_{s \to \infty} \frac{\int_s^\infty \|\gamma'(\tau)\| \, \mathrm{d}\tau}{\|z - \gamma(s)\|} = 1,$$

which implies that $\gamma$ has finite length. $\qquad\square$

The following Curve Selection Lemma is crucial in proving the Kurdyka-Łojasiewicz inequalities.

**Lemma B.6** (Curve Selection [Kurdyka, 1998, Proposition 1]). *Given a definable set $A \in \mathbb{R}^n$ and $x \in \overline{A \setminus \{x\}}$, there exists a definable curve $\gamma : [0, 1] \to \mathbb{R}^n$ which is $C^1$ on $[0, 1]$ and satisfies $\gamma(0) = x$ and $\gamma\left((0, 1]\right) \subset A \setminus \{x\}$.*

We also need the following version at infinity, from [Némethi and Zaharia, 1992, Lemma 2] and [Kurdyka et al., 2000b, Lemma 3.4].

**Lemma B.7** (Curve Selection at Infinity). *Given a definable set $A \in \mathbb{R}^n$, a definable function $f : A \to \mathbb{R}$, and a sequence $x_i$ in $A$ such that $\lim_{i \to \infty} \|x_i\| = \infty$ and $\lim_{i \to \infty} f(x_i) = y$, there exists a positive constant $a$ and a $C^1$ definable curve $\rho : [a, \infty) \to A$ such that $\|\rho(s)\| = s$, and $\lim_{s \to \infty} f\left(\rho(s)\right) = y$.*

*Proof.* For any $x \in \mathbb{R}^n$, let $x(j)$ denote the $j$-th coordinate of $x$, and consider the definable map $\psi : A \to \mathbb{R}^{n+2}$ given by

$$\psi(x) := \left( \frac{x(1)}{\sqrt{1 + \|x\|^2}}, \ldots, \frac{x(n)}{\sqrt{1 + \|x\|^2}}, \frac{1}{\sqrt{1 + \|x\|^2}}, f(x) \right).$$

By construction, the first $n + 1$ coordinates of $\psi(x)$ are bounded for all $x$; since furthermore $\lim_{i \to \infty} f(x_i) = y$ with $\lim_{i \to \infty} \|x_i\| \to \infty$, then $\psi$ has an accumulation point $(u, 0, y)$ for some $\|u\| = 1$, where $(u, 0, y) \in \overline{\psi(A)} \setminus \{(u, 0, y)\}$. We can therefore apply Lemma B.6, obtaining a $C^1$ definable curve $\gamma : [0, 1] \to \mathbb{R}^{n+2}$ such that $\gamma(0) = (u, 0, y)$ and $\gamma\big((0, 1]\big) \subset \psi(A)$.

With this in hand, define a curve $\rho_0 : [1, \infty) \to A$ as

$$\rho_0(s) := \psi^{-1}\left( \gamma\left(\frac{1}{s}\right) \right),$$

which is $C^1$ definable and satisfies $\lim_{s \to \infty} \|\rho_0(s)\| = \infty$ and $\lim_{s \to \infty} f\big(\rho_0(s)\big) = y$. Theorem B.4 implies that $\mathrm{d}\|\rho_0(s)\| / \mathrm{d}s$ is positive and continuous for all large enough $s$; to finish the proof, we may obtain a $C^1$ definable $\rho$ from $\rho_0$ via reparameterization (i.e., composing $\rho_0$ with some other $C^1$ definable function from $\mathbb{R}$ to $\mathbb{R}$) so that $\|\rho(s)\| = s$ on $[a, \infty)$ for some $a \in \mathbb{R}$. $\qquad\square$

## B.2  Clarke subdifferentials

Here we prove the definability of Clarke subdifferential, and a *chain rule along arcs* which is crucial in our analysis.

Here is a standard result on the definability of (Fréchet) derivatives: given a definable function $f : D \to \mathbb{R}$ with an open domain $D$, the set

$$\left\{ (x, x^*) \big| \, f \text{ is Fréchet differentiable at } x, \nabla f(x) = x^* \right\}$$

is definable, since it is given by the following first-order formula:

$$(x, x^*) : \quad x \in D \ \wedge$$
$$\forall \epsilon > 0 \exists \delta > 0 \forall x' \in D \left( (\|x - x'\| < \delta) \Rightarrow f(x') - f(x) - \langle x^*, x' - x \rangle < \epsilon \|x - x'\| \right).$$

Now consider a locally Lipschitz definable function $f : D \to \mathbb{R}$ with an open domain $D$. Local Lipschitz continuity ensures that Gâteaux and Fréchet differentiability coincide [Borwein and Lewis, 2000, Exercise 6.2.5], and $f$ is differentiable a.e. [Borwein and Lewis, 2000, Theorem 9.1.2]. Recall that the Clarke subdifferential at $x \in D$ is defined as

$$\partial f(x) := \mathrm{conv} \left\{ \lim_{i \to \infty} \nabla f(x_i) \middle| x_i \in D, \nabla f(x_i) \text{ exists}, \lim_{i \to \infty} x_i = x \right\},$$

and that $\bar{\partial} f(x)$ denotes the unique minimum-norm subgradient. Similarly to the gradients, the following result holds for the Clarke subdifferentials.

**Lemma B.8.** *Given a locally Lipschitz definable function $f : D \to \mathbb{R}$ with an open domain $D \subset \mathbb{R}^n$, the set*

$$\Gamma := \left\{ (x, x^*) \big| \, x \in D, x^* \in \partial f(x) \right\}$$

*is definable. Moreover, the function $D \ni x \mapsto \bar{\partial} f(x)$ is definable.*

*Proof.* Let $D' := \left\{ x \in D \big| \nabla f(x) \text{ exists} \right\}$, which is definable. The set $A$ given by

$$(x, y) : \quad x \in D \ \wedge \ \forall \epsilon > 0 \exists x' \in D' \left( \|x - x'\| < \epsilon \right) \wedge \left( \|y - \nabla f(x')\| < \epsilon \right)$$

is also definable. Now by Carathéodory's Theorem, $\Gamma$ is given by

$$(x, x^*) : \quad \exists (x_1, x_1^*), \ldots, (x_{n+1}, x_{n+1}^*) \in A \exists \lambda_1, \ldots, \lambda_{n+1} \geq 0$$
$$(x_1 = x) \wedge \cdots \wedge (x_{n+1} = x) \wedge \left( \sum_{i=1}^{n+1} \lambda_i = 1 \right) \wedge \left( \sum_{i=1}^{n+1} \lambda_i x_i^* = x^* \right).$$

It then follows from Lemma B.1 that $x \mapsto \|\bar{\partial} f(x)\|$ and $x \mapsto \bar{\partial} f(x)$ are definable. $\qquad\square$

The following *chain rule* is important in our analysis; it allows us to use $\bar{\partial}f$ in many places that seem to call on $\nabla f$. It is basically from [Davis et al., 2020, Theorem 5.8 and Lemma 5.2], though we detail how their proof handles our slight extension.

**Lemma B.9.** *Given a locally Lipschitz definable $f : D \to \mathbb{R}$ with an open domain $D$, for any interval $I$ and any arc $z : I \to D$, it holds for a.e. $t \in I$ that*

$$\frac{\mathrm{d}f(z_t)}{\mathrm{d}t} = \left\langle z_t^*, \frac{\mathrm{d}z_t}{\mathrm{d}t} \right\rangle, \quad \textit{for all } z_t^* \in \partial f(z_t).$$

*Moreover, for the gradient flow in eq. (1), it holds for a.e. $t \geq 0$ that $\mathrm{d}W_t / \mathrm{d}t = -\bar{\partial}\mathcal{L}(W_t)$ and $\mathrm{d}\mathcal{L}(W_t) / \mathrm{d}t = -\left\|\bar{\partial}\mathcal{L}(W_t)\right\|^2$.*

*Proof.* The first part is proved in [Davis et al., 2020, Theorem 5.8] when $D = \mathbb{R}^n$ and $I = [0, \infty)$, but actually holds in general as verified below. Note that for any $t \in I$ excluding the endpoints, since $f$ is locally Lipschitz, there exists a neighborhood $U$ of $z(t)$ on which $f$ is $K$-Lipschitz continuous. Let $g$ denote the infimal convolution of $f|_U$ and $K\|\cdot\|$. It follows that $g$ is definable (Lemma B.1) and $K$-Lipschitz continuous on $\mathbb{R}^n$, and $f = g$ on $U$ [Borwein and Lewis, 2000, Exercise 7.1.2]. Take an interval $[a, b] \ni t$ with rational endpoints such that $z([a, b]) \subset U$, and define the absolutely continuous curve $\tilde{z} : [0, \infty) \to D$ as $\tilde{z}(t) = z(a + t)$ for $t \in [0, b-a]$, and $\tilde{z}(t) = z(b)$ for $t > b - a$. Applying [Davis et al., 2020, Theorem 5.8] to $g$ and $\tilde{z}$ gives that the chain rule holds for $f$ and $z$ a.e. on $[a, b]$. Since this holds for any $t \in I$, and there are only countably many intervals with rational endpoints, it follows that the chain rule holds a.e. for $f$ and $z$ on $I$. The second claim of Lemma B.9 can be proved in the same way as [Davis et al., 2020, Lemma 5.2]. $\square$

## B.3 Kurdyka-Łojasiewicz inequalities

**Asymptotic Clarke critical values.** To prove the Kurdyka-Łojasiewicz inequalities, we need the notion of asymptotic Clarke critical values, introduced in [Bolte et al., 2007]. Given a locally Lipschitz function $f : D \to \mathbb{R}$ with an open domain $D$, we say that $a \in \mathbb{R} \cup \{+\infty, -\infty\}$ is an asymptotic Clarke critical value of $f$ if there exists a sequence $(x_i, x_i^*)$ where $x_i \in D$ and $x_i^* \in \partial f(x_i)$, such that $\lim_{i \to \infty}(1 + \|x_i\|)\|x_i^*\| = 0$ and $\lim_{i \to \infty} f(x_i) = a$.

We have the following result regarding the asymptotic Clarke critical values of a definable function, which is basically from [Bolte et al., 2007, Corollary 9].

**Lemma B.10.** *Given a locally Lipschitz definable function $f : D \to \mathbb{R}$ with an open domain $D$, it holds that $f$ has finitely many asymptotic Clarke critical values.*

To state the proof in a bit more detail, [Bolte et al., 2007, Corollary 9] shows that if $f$ is lower semi-continuous and $f > -\infty$, then $f$ has finitely many asymptotic Clarke critical values. To get Lemma B.10, we just need to apply [Bolte et al., 2007, Corollary 9] to the lower semi-continuous envelopes of $f|_{f^{-1}((0,\infty))}$ and $-f|_{f^{-1}((-\infty,0))}$.

**The bounded setting.** Here we consider the case where the domain of $f$ is bounded. [Kurdyka, 1998, Theorem 1] gives a Kurdyka-Łojasiewicz inequality assuming $f$ is differentiable; below we extend it to the locally Lipschitz setting.

**Lemma B.11.** *Given a locally Lipschitz definable function $f : D \to \mathbb{R}$ with an open bounded domain $D$, there exists $\nu > 0$ and a definable desingularizing function $\Psi$ on $[0, \nu)$ such that*

$$\Psi'\left(f(x)\right)\left\|\bar{\partial}f(x)\right\| \geq 1$$

*for any $x \in f^{-1}\left((0, \nu)\right)$.*

*Proof.* Since $f$ is definable, $f(D)$ is also definable, and thus is a finite union of open intervals and points. It follows that either there exists $\epsilon > 0$ such that $(0, \epsilon) \cap f(D) = \emptyset$, in which case the claim trivially holds; otherwise we are free to choose $\epsilon > 0$ such that $(0, \epsilon) \subset f(D)$. In the second case, define $\phi : (0, \epsilon) \to \mathbb{R}$ as

$$\phi(z) := \inf\left\{\left\|\bar{\partial}f(x)\right\| \,\middle|\, f(x) = z\right\}.$$

By Lemmas B.1 and B.8, $\phi$ is definable. Lemma B.10 implies that there are only finitely many asymptotic Clarke critical values on $(0, \epsilon)$, and thus there exists $\epsilon' \in (0, \epsilon)$ such that on $(0, \epsilon')$ there is no asymptotic Clarke critical value and $\phi(z) > 0$.

Now consider the definable set

$$A := \left\{ x \in f^{-1}\left((0, \epsilon')\right) \middle| \left\|\bar{\partial} f(x)\right\| \leq 2\phi\left(f(x)\right) \right\}.$$

It follows that there exists a sequence $x_i$ in $A$ such that $f(x_i) \to 0$. Since the domain of $f$ is bounded, $x_i$ has an accumulation point $y$. Applying Lemma B.6 to the graph of $f|_A$, we have that there exists a $C^1$ definable curve $(\rho, h) : [0, 1] \to \mathbb{R}^{n+1}$ such that $\rho(0) = y$, and $h(0) = 0$, and $\rho\left((0, 1]\right) \subset A$, and $h(s) = f\left(\rho(s)\right)$ on $(0, 1]$.

1. Since $\rho$ is $C^1$ on $[0, 1]$, there exists $B > 0$ such that $\left\|\rho'(s)\right\| \leq B$ on $[0, 1]$.

2. Since $h$ is definable, $h(0) = 0$, and $h(s) > 0$ on $(0, 1]$, Theorem B.4 implies that there exists a constant $\omega \in (0, 1]$ such that $h'(s) > 0$ on $(0, \omega)$.

3. Lemma B.9 implies that for a.e. $s \in (0, \omega)$,

$$h'(s) - \left\langle \bar{\partial} f\left(\rho(s)\right), \rho'(s) \right\rangle = 0. \tag{6}$$

Since the left hand side of eq. (6) is definable, it can actually be nonzero only for finitely many $s$, and thus is equal to 0 on some interval $(0, \mu)$ where $\mu \leq \omega$.

4. Let $\nu = h(\mu)$, the Inverse Function Theorem implies that $\Psi : (0, \nu) \to (0, 2B\mu)$ given by $\Psi(z) := 2Bh^{-1}(z)$ is also $C^1$ definable with a positive derivative, and $\lim_{z \to 0} \Psi(z) = 0$.

Now for any $x \in f^{-1}\left((0, \nu)\right)$, let $s = h^{-1}\left(f(x)\right)$, we have

$$
\begin{aligned}
\Psi'\left(f(x)\right)\left\|\bar{\partial} f(x)\right\| &= \frac{2B}{h'(s)}\left\|\bar{\partial} f(x)\right\| && \text{(Inverse Function Theorem)} \\
&\geq \frac{2B}{h'(s)} \cdot \frac{1}{2}\left\|\bar{\partial} f\left(\rho(s)\right)\right\| && \text{(Definition of } A\text{)} \\
&= \frac{B\left\|\bar{\partial} f\left(\rho(s)\right)\right\|}{\left\langle \bar{\partial} f\left(\rho(s)\right), \rho'(s) \right\rangle} \geq 1. && \text{(Bullet 3 above \& Cauchy-Schwarz)}
\end{aligned}
$$

$\square$

**The unbounded setting.** The unbounded setting is more complicated: to show directional convergence, we need two Kurdyka-Łojasiewicz inequalities (cf. Lemmas 3.5 and 3.6), depending on the relationship between the spherical and radial parts of $\bar{\partial} f$.

Given a locally Lipschitz definable function $f : D \to \mathbb{R}$ with an open domain $D \subset \left\{x \middle| \|x\| > 1\right\}$, recall that $\bar{\partial}_r f(x)$ and $\bar{\partial}_\perp f(x)$ denote the radial part and spherical part of $\bar{\partial} f(x)$ respectively, which are both definable. Given $\epsilon, c, \eta > 0$, let

$$U_{\epsilon, c, \eta} := \left\{ x \in D \middle| f(x) \in (0, \epsilon), \left\|\bar{\partial}_\perp f(x)\right\| \geq c\|x\|^\eta\left\|\bar{\partial}_r f(x)\right\| \right\}.$$

In any o-minimal structure, $U_{\epsilon, c, \eta}$ is definable if $\eta$ is rational. Now we prove Lemma 3.5, a Kurdyka-Łojasiewicz inequality on some $U_{\nu, c, \eta}$, using ideas from [Kurdyka et al., 2006, Proposition 6.3].

*Proof of Lemma 3.5.* Similarly to the proof of Lemma B.11, we only need to consider the case where there exists $\epsilon > 0$ such that $(0, \epsilon) \subset f(D)$. Without loss of generality, we can assume $\eta$ is rational, since otherwise we can consider any rational $\eta' \in (0, \eta)$. Therefore $U_{\epsilon, c, \eta}$ is definable, and so is $f(U_{\epsilon, c, \eta})$. If there exists $\epsilon' > 0$ such that $f(U_{\epsilon, c, \eta}) \cap (0, \epsilon') = \emptyset$, then Lemma 3.5 trivially holds; therefore we assume that there exists $\epsilon' > 0$ such that $f(U_{\epsilon', c, \eta}) = (0, \epsilon')$. By Lemma B.10, we

can also make $\epsilon'$ small enough so that there is no asymptotic Clarke critical value on $(0, \epsilon')$. Define $\phi : (0, \epsilon') \to \mathbb{R}$ as

$$\phi(z) := \inf \left\{ \|x\| \|\bar{\partial} f(x)\| \,\middle|\, x \in U_{\epsilon', c, \eta}, f(x) = z \right\}.$$

Since there is no asymptotic Clarke critical value on $(0, \epsilon')$, it holds that $\phi(z) > 0$.

Consider the definable set

$$A := \left\{ x \in U_{\epsilon', c, \eta} \,\middle|\, \|x\| \|\bar{\partial} f(x)\| \leq 2\phi\left(f(x)\right) \right\}.$$

Since $f(U_{\epsilon', c, \eta}) = (0, \epsilon')$ as above, there exists a sequence $x_i$ in $A$ such that $f(x_i) \to 0$. If the $x_i$ are bounded, then the claim follows from the proof of Lemma B.11 and $D \subset \{x \mid \|x\| > 1\}$. If the $x_i$ are unbounded, then without loss of generality (e.g., by taking a subsequence) we can assume $\|x_i\| \to \infty$. Lemma B.7 asserts that there exists a $C^1$ definable curve $\rho : [a, \infty) \to A$ such that $\|\rho(s)\| = s$ and $\lim_{s \to \infty} f\left(\rho(s)\right) = 0$. Let $h(s) := f\left(\rho(s)\right)$, and $\rho'_r(s) := \langle \rho'(s), \rho(s) \rangle \rho(s)/s^2$ denote the radial part of $\rho'(s)$, and $\rho'_\perp(s) := \rho'(s) - \rho'_r(s)$ denote the spherical part of $\rho'(s)$.

1. Theorem B.4 implies that $h'$ is negative and continuous on some interval $[\omega, \infty)$.

2. As in the proof of Lemma B.11, it follows from Lemma B.9 that there exists $\mu \geq \omega$, such that

$$h'(s) - \left\langle \bar{\partial} f\left(\rho(s)\right), \rho'(s) \right\rangle = 0$$

   for all $s \in [\mu, \infty)$.

3. Note that for all $s \in [\mu, \infty)$,

$$\left| h'(s) \right| = \left| \left\langle \bar{\partial} f\left(\rho(s)\right), \rho'(s) \right\rangle \right| = \left| \left\langle \bar{\partial}_r f\left(\rho(s)\right), \rho'_r(s) \right\rangle + \left\langle \bar{\partial}_\perp f\left(\rho(s)\right), \rho'_\perp(s) \right\rangle \right|$$

$$\leq \left\| \bar{\partial}_r f\left(\rho(s)\right) \right\| + \left\| \bar{\partial}_\perp f\left(\rho(s)\right) \right\| \|\rho'_\perp(s)\|$$

$$\leq \left( \frac{1}{cs^\eta} + \|\rho'_\perp(s)\| \right) \left\| \bar{\partial}_\perp f\left(\rho(s)\right) \right\|$$

   since $\|\rho'_r(s)\| = 1$ and $\rho([a, \infty)) \subset U_{\epsilon', c, \eta}$. Let $\tilde{\rho}(s) := \rho(s)/s$, we have

$$\frac{\mathrm{d}\tilde{\rho}(s)}{\mathrm{d}s} = \frac{\rho'_\perp(s)}{s}.$$

   Since $\tilde{\rho}(s)$ is a $C^1$ definable curve on the unit sphere, Proposition B.3 and Lemma B.5 imply that $\|\rho'_\perp(s)\| / s$ is integrable on $[\mu, \infty)$. Therefore

$$\left| h'(s) \right| \leq -g'(s) \cdot s \left\| \bar{\partial}_\perp f\left(\rho(s)\right) \right\|,$$

   where

$$g(s) := \int_s^\infty \left( \frac{1}{c\tau^{1+\eta}} + \frac{\|\rho'_\perp(\tau)\|}{\tau} \right) \mathrm{d}\tau.$$

Let $\nu = h(\mu)$, and define $\Psi : (0, \nu) \to \mathbb{R}$ as

$$\Psi(z) := 2g\left(h^{-1}(z)\right).$$

It holds that $\lim_{z \to 0} \Psi(z) = 0$. Moreover, for any $x \in U_{\nu, c, \eta}$, let $s = h^{-1}\left(f(x)\right)$, we have

$$\Psi'\left(f(x)\right) \|x\| \|\bar{\partial} f(x)\| = \frac{2g'(s)}{h'(s)} \|x\| \|\bar{\partial} f(x)\| \geq \frac{2g'(s)}{h'(s)} \cdot \frac{1}{2} s \left\| \bar{\partial} f\left(\rho(s)\right) \right\| \geq 1.$$

$\square$

Below we prove Lemma 3.6, a Kurdyka-Łojasiewicz inequality which is useful outside of $U_{\nu,c,\eta}$.

*Proof of Lemma 3.6.* We first assume that $\lambda$ is rational, and later finish by handling the real case with a quick reduction. Consider the definable mapping $\xi_\lambda : \mathbb{R}^n \setminus \{0\} \to \mathbb{R}^n \setminus \{0\}$ given by

$$\xi_\lambda(x) := \frac{x}{\|x\|^{1+\lambda}}.$$

Note that $\xi_\lambda^{-1} = \xi_{1/\lambda}$. If $y = \xi_\lambda(x)$, then $x = \xi_{1/\lambda}(y)$, which has the Jacobian

$$
\begin{aligned}
\frac{\partial(x_1, \ldots, x_n)}{\partial(y_1, \ldots, y_n)} &= \frac{\partial \xi_{1/\lambda}(y)}{\partial(y_1, \ldots, y_n)} \\
&= \|y\|^{-(1+\lambda)/\lambda} \left( I - \frac{1+\lambda}{\lambda} \frac{y}{\|y\|} \frac{y^\top}{\|y\|} \right) \\
&= \|x\|^{1+\lambda} \left( I - \frac{1+\lambda}{\lambda} \frac{x}{\|x\|} \frac{x^\top}{\|x\|} \right).
\end{aligned}
\tag{7}
$$

Define $g : \xi_\lambda(D) \to \mathbb{R}$ as

$$g(y) := f\left(\xi_\lambda^{-1}(y)\right).$$

Note that $g$ is locally Lipschitz and definable with an open bounded domain. Therefore Lemma B.11 implies that there exists $\nu > 0$ and a definable desingularizing function $\Psi$ on $[0, \nu)$ such that

$$\Psi'\left(g(y)\right) \left\| \bar{\partial} g(y) \right\| \geq 1$$

for any $y \in g^{-1}\left((0, \nu)\right)$. Let $x = \xi_\lambda^{-1}(y)$, it holds that $g$ is differentiable at $y$ if and only if $f$ is differentiable at $x$, and by the definition of Clarke subdifferential,

$$y^* := \left( \frac{\partial(x_1, \ldots, x_n)}{\partial(y_1, \ldots, y_n)} \right)^\top \bar{\partial} f(x) \in \partial g(y).$$

Therefore eq. (7) implies that

$$\left\| \bar{\partial} g(y) \right\| \leq \|y^*\| = \|x\|^{1+\lambda} \left\| \bar{\partial}_\perp f(x) - \frac{1}{\lambda} \bar{\partial}_r f(x) \right\| \leq \max\left\{ 1, \frac{1}{\lambda} \right\} \|x\|^{1+\lambda} \left\| \bar{\partial} f(x) \right\|,$$

and thus

$$\max\left\{ 1, \frac{1}{\lambda} \right\} \Psi'\left(f(x)\right) \|x\|^{1+\lambda} \left\| \bar{\partial} f(x) \right\| \geq 1,$$

which finishes the proof for rational $\lambda$. To handle real $\lambda > 0$, we can apply the above result to any rational $\lambda' \in (\lambda/2, \lambda)$. $\qquad\square$

## C  Omitted proofs from Section 3

We first give a generalization of Euler's homogeneous function theorem, which can also be found in [Lyu and Li, 2019, Theorem B.2], but with an additional requirement of a chain rule.

**Lemma C.1.** *Suppose $f : \mathbb{R}^n \to \mathbb{R}$ is locally Lipschitz and $L$-positively homogeneous for some $L > 0$, then for any $x \in \mathbb{R}^n$ and any $x^* \in \partial f(x)$,*

$$\langle x, x^* \rangle = L f(x).$$

*Proof.* Let $D'$ denote the set of $x$ where $f$ is differentiable. For any nonzero $x \in D'$, it holds that

$$\lim_{\delta \downarrow 0} \frac{f(x + \delta x) - f(x) - \langle \nabla f(x), \delta x \rangle}{\delta \|x\|} = 0.$$

Since $f$ is $L$-positively homogeneous, $f(x + \delta x) = (1 + \delta)^L f(x)$, and thus

$$\lim_{\delta \downarrow 0} \frac{\left((1+\delta)^L - 1\right) f(x) - \langle \nabla f(x), \delta x \rangle}{\delta \|x\|} = 0,$$

which implies $\langle x, \nabla f(x) \rangle = Lf(x)$. This property trivially holds if $0 \in D'$.

Now consider an arbitrary $x \in \mathbb{R}^n$. For any sequence $x_i$ in $D'$ such that $\lim_{i \to \infty} x_i = x$ and $\lim_{i \to \infty} \nabla f(x_i) = x^*$, it holds that

$$\langle x, x^* \rangle = \lim_{i \to \infty} \langle x_i, \nabla f(x_i) \rangle = \lim_{i \to \infty} Lf(x_i) = Lf(x).$$

Since $\partial f(x)$ consists of convex combinations of such $x^*$, Lemma C.1 holds. $\qquad\square$

Next we prove a few technical lemmas. Recall the definitions of unnormalized and normalized smoothed margin: given $W \neq 0$, let

$$\alpha(W) := \ell^{-1}\left(\mathcal{L}(W)\right), \quad \text{and} \quad \tilde{\alpha}(W) := \frac{\alpha(W)}{\|W\|^L}.$$

Additionally, given any function $f$ which is locally Lipschitz around a nonzero $W$, let

$$\bar{\partial}_r f(W) := \left\langle \bar{\partial} f(W), \widetilde{W} \right\rangle \widetilde{W} \quad \text{and} \quad \bar{\partial}_\perp f(W) := \bar{\partial} f(W) - \bar{\partial}_r f(W)$$

denote the radial and spherical parts of $\bar{\partial} f(W)$ respectively.

We first characterize the Clarke subdifferentials of $\alpha$, the unnormalized smoothed margin.

**Lemma C.2.** *It holds for any $W \in \mathbb{R}^k$ that*

$$\bar{\partial}\alpha(W) = \frac{\bar{\partial}\mathcal{L}(W)}{\ell'\left(\alpha(W)\right)}, \quad \text{and } \beta(W) := \frac{\langle W, \bar{\partial}\alpha(W) \rangle}{L} = \frac{\langle W, W^* \rangle}{L} \text{ for any } W^* \in \partial\alpha(W).$$

*Proof.* Note that $\mathcal{L}$ is differentiable at $W$ if and only if $\alpha$ is differentiable at $W$, and when both gradients exist, the chain rule and inverse function theorem together imply that

$$\nabla\alpha(W) = \frac{\nabla\mathcal{L}(W)}{\ell'\left(\ell^{-1}\left(\mathcal{L}(W)\right)\right)} = \frac{\nabla\mathcal{L}(W)}{\ell'\left(\alpha(W)\right)},$$

whereby the first claim follows from the definition of Clarke subdifferential. To prove the second claim, the chain rule for Clarke subdifferentials [Clarke, 1983, Theorem 2.3.9] implies that

$$\partial\alpha(W) \subset \text{conv}\left(\sum_{i=1}^n \frac{\ell'\left(p_i(W)\right)}{\ell'\left(\alpha(W)\right)} \partial p_i(W)\right),$$

and thus Lemma C.1 ensures for any $W^* \in \partial\alpha(W)$,

$$\frac{\langle W, W^* \rangle}{L} = \sum_{i=1}^n \frac{\ell'\left(p_i(W)\right)}{\ell'\left(\alpha(W)\right)} p_i(W) = \beta(W),$$

which finishes the proof. $\qquad\square$

Next we note that the Clarke subdifferentials of $\alpha$ and $\tilde{\alpha}$ are strongly related.

**Lemma C.3.** *For any nonzero $W \in \mathbb{R}^k$, we have*

$$\bar{\partial}_r\tilde{\alpha}(W) = L\frac{\beta(W) - \alpha(W)}{\|W\|^{L+1}}\widetilde{W}, \quad \text{and} \quad \bar{\partial}_\perp\tilde{\alpha}(W) = \frac{\bar{\partial}_\perp\alpha(W)}{\|W\|^L}.$$

*Proof.* Note that given $W \neq 0$, $\alpha$ is differentiable at $W$ if and only if $\tilde{\alpha}$ is differentiable at $W$, and when both gradients exist,

$$\nabla\tilde{\alpha}(W) = \frac{\nabla\alpha(W)}{\|W\|^L} - \frac{\alpha(W) \cdot L\|W\|^{L-1}\widetilde{W}}{\|W\|^{2L}} = \frac{\nabla\alpha(W)}{\|W\|^L} - L\frac{\alpha(W)\widetilde{W}}{\|W\|^{L+1}}.$$

By the definition of Clarke subdifferential, for any nonzero $W$,

$$\partial\tilde{\alpha}(W) = \left\{ \frac{W^*}{\|W\|^L} - L\frac{\alpha(W)\widetilde{W}}{\|W\|^{L+1}} \,\middle|\, W^* \in \partial\alpha(W) \right\}. \tag{8}$$

The first claim of Lemma C.3 holds since for any $W \in \partial\alpha(W)$, by Lemma C.2,

$$\left\langle \frac{W^*}{\|W\|^L} - L\frac{\alpha(W)\widetilde{W}}{\|W\|^{L+1}}, \widetilde{W} \right\rangle = L\frac{\beta(W)}{\|W\|^{L+1}} - L\frac{\alpha(W)}{\|W\|^{L+1}}.$$

To prove the second claim, note that since $\partial\alpha(W)$ and $\partial\tilde{\alpha}(W)$ have fixed radial parts, the norms of the whole subgradients are minimized if and only if the norms of their spherical parts are minimized. Due to eq. (8), the norms of the spherical parts of $\partial\alpha(W)$ and $\partial\tilde{\alpha}(W)$ are minimized simultaneously, and the second claim follows. $\qquad\square$

The last technical result we need is that $\alpha$ and $\beta$ are close.

**Lemma C.4.** *For $\ell \in \{\ell_{\exp}, \ell_{\log}\}$ and any $W$ satisfying $\mathcal{L}(W) < \ell(0)$, it holds that*

$$0 < \alpha(W) \leq \beta(W) \leq \alpha(W) + 2\ln(n) + 1.$$

To prove Lemma C.4, we need the following result on $\ell_{\exp}$ and $\ell_{\log}$. Define $\sigma : \mathbb{R}_+ \to \mathbb{R}$ by

$$\sigma(z) := \ell'\left(\ell^{-1}(z)\right)\ell^{-1}(z), \tag{9}$$

and $\pi : \mathbb{R}^n \to \mathbb{R}$ by

$$\pi(v) := \ell^{-1}\left(\sum_{i=1}^n \ell(v_i)\right). \tag{10}$$

Note that $\alpha(W) = \pi\left(p(W)\right)$ where $p(W) = \left(p_1(W), \ldots, p_n(W)\right)$.

**Lemma C.5.** *For $\ell \in \{\ell_{\exp}, \ell_{\log}\}$, it holds that $\sigma$ is super-additive on $\left(0, \ell(0)\right)$, meaning that $\sigma(z_1 + z_2) \geq \sigma(z_1) + \sigma(z_2)$ for any $z_1, z_2 > 0$ such that $z_1 + z_2 < \ell(0)$. Moreover $\pi$ is concave.*

*Proof.* For $\ell_{\exp}(z) = e^{-z}$, we have $\sigma(z) = z\ln(z)$, while for $\ell_{\log}(z) = \ln(1 + e^{-z})$, we have $\sigma(z) = (1 - e^{-z})\ln(e^z - 1)$. In both cases $\lim_{z\to 0}\sigma(z) = 0$, and $\sigma$ is convex on $\left(0, \ell(0)\right)$, which implies super-additivity.

Turning to concavity of $\pi$, in the case of $\ell_{\exp}$, it is a standard fact in convex analysis that the function $\pi(v) = -\ln\sum_{i=1}^n \exp(-v_i)$ is concave [Borwein and Lewis, 2000, Exercise 3.3.7]. For $\ell_{\log}$, note that

$$\frac{\partial\pi}{\partial v_i} = \frac{\ell'(v_i)}{\ell'\left(\ell^{-1}\left(\sum_{i=1}^n \ell(v_i)\right)\right)} = \frac{\ell'(v_i)}{\exp\left(-S(v)\right) - 1},$$

where $S(v) := \sum_{i=1}^n \ell(v_i)$, and

$$\nabla^2\pi(v) = \frac{1}{\exp\left(-S(v)\right) - 1}\operatorname{diag}\left(\ell''(v_1), \ldots, \ell''(v_n)\right) + \frac{\exp\left(-S(v)\right)}{\left(\exp\left(-S(v)\right) - 1\right)^2}\nabla S(v)\nabla S(v)^{\intercal}.$$

We want to show that $\nabla^2\pi(v) \preceq 0$, or equivalently

$$\left(\exp\left(S(v)\right) - 1\right)\operatorname{diag}\left(\ell''(v_1), \ldots, \ell''(v_n)\right) - \nabla S(v)\nabla S(v)^{\intercal} \succeq 0.$$

By definition, we need to show that for any $z \in \mathbb{R}^n$,

$$\left(\exp\left(S(v)\right) - 1\right)\sum_{i=1}^n \ell''(v_i)z_i^2 \geq \left(\sum_{i=1}^n \ell'(v_i)z_i\right)^2.$$

Note that for $a, b > 0$, we have $e^{a+b} - 1 > (e^a - 1) + (e^b - 1)$, which implies

$$\exp\big(S(v)\big) - 1 > \sum_{i=1}^{n} \Big(\exp\big(\ell(v_i)\big) - 1\Big) = \sum_{i=1}^{n} e^{-v_i}.$$

Also note that $e^{-v_i} \ell''(v_i) = \ell'(v_i)^2$, and thus

$$\Big(\exp\big(S(v)\big) - 1\Big) \sum_{i=1}^{n} \ell''(v_i) z_i^2 \geq \sum_{i=1}^{n} e^{-v_i} \sum_{i=1}^{n} \ell''(v_i) z_i^2 \geq \left(\sum_{i=1}^{n} \ell'(v_i) z_i\right)^2.$$

$\square$

Using Lemma C.5, we can prove Lemma C.4.

*Proof of Lemma C.4.* For simplicity, let $p := \big(p_1(W), \ldots, p_n(W)\big)$. Recall that $\alpha(W) = \pi(p)$, and from the proof of Lemma C.2 we know that

$$\beta(W) = \sum_{i=1}^{n} \frac{\ell'\big(p_i(W)\big)}{\ell'\big(\ell^{-1}\big(\mathcal{L}(W)\big)\big)} p_i(W) = \big\langle \nabla \pi(p), p \big\rangle.$$

By the super-additivity of the function $\sigma$ defined in eq. (9), we know that

$$\begin{aligned}
\sum_{i=1}^{n} \ell'\big(p_i(W)\big) p_i(W) &= \sum_{i=1}^{n} \ell'\Big(\ell^{-1}\big(\ell(p_i(W))\big)\Big) \ell^{-1}\big(\ell(p_i(W))\big) \\
&\leq \ell'\Big(\ell^{-1}\big(\mathcal{L}(W)\big)\Big) \ell^{-1}\big(\mathcal{L}(W)\big) \\
&= \ell'\Big(\ell^{-1}\big(\mathcal{L}(W)\big)\Big) \alpha(W),
\end{aligned}$$

and since $\ell' < 0$, we have $\beta(W) \geq \alpha(W)$.

On the other claim, for $\ell_{\exp}$, since $\pi$ is concave,

$$\beta(W) = \big\langle \nabla \pi(p), p \big\rangle = \big\langle \nabla \pi(p), p - 0 \big\rangle \leq \pi(p) - \pi(0) = \alpha(W) + \ln(n).$$

For $\ell_{\log}$, note that on the interval $\big(0, \ell(0)\big)$, the function $h(z) := \ell'\big(\ell^{-1}(z)\big) = e^{-z} - 1$ is convex with $\lim_{z \to 0} h(z) = 0$ and $h'(z) \in (-1, -1/2)$, and thus

$$\big\|\pi(p)\big\|_1 = \sum_{i=1}^{n} \frac{\ell'\big(p_i(W)\big)}{\ell'\big(\ell^{-1}\big(\mathcal{L}(W)\big)\big)} \leq 2.$$

Let $c = -\ln\Big(\exp\big(\ln(2)/n\big) - 1\Big) \leq \ln(n) - \ln\ln(2)$ and $\vec{1}$ denote the all-ones vector, we have $\pi\big(c\vec{1}\big) = 0$, and

$$\begin{aligned}
\beta(W) = \big\langle \nabla \pi(p), p \big\rangle = \Big\langle \nabla \pi(p), p - c\vec{1} \Big\rangle + \Big\langle \nabla \pi(p), c\vec{1} \Big\rangle \\
\leq \pi(p) - \pi\big(c\vec{1}\big) + c\big\|\pi(p)\big\|_1 \\
= \alpha(W) + c\big\|\pi(p)\big\|_1 \\
\leq \alpha(W) + 2\ln(n) - 2\ln\ln(2) \leq \alpha(W) + 2\ln(n) + 1.
\end{aligned}$$

$\square$

Now we can prove Lemma 3.4.

*Proof of Lemma 3.4.* Lemma B.9 implies that for a.e. $t \geq 0$,

$$\frac{dW_t}{dt} = -\bar{\partial}\mathcal{L}(W_t).$$

First note that Assumption 2.2 implies that $\|W_0\| > 0$, and moreover Lyu and Li [2019, Lemma 5.1] proved that $d\|W_t\|/dt > 0$ for a.e. $t \geq 0$, and thus $\|W_t\|$ is increasing and $\|W_t\| \geq \|W_0\| > 0$.

Now we have for a.e. $t \geq 0$,

$$\frac{d\tilde{\alpha}(W_t)}{dt} = \left\langle \bar{\partial}\tilde{\alpha}(W_t), -\bar{\partial}\mathcal{L}(W_t) \right\rangle = \left\langle \bar{\partial}_r\tilde{\alpha}(W_t), -\bar{\partial}_r\mathcal{L}(W_t) \right\rangle + \left\langle \bar{\partial}_\perp\tilde{\alpha}(W_t), -\bar{\partial}_\perp\mathcal{L}(W_t) \right\rangle.$$

By Lemmas C.2 to C.4, both $\left\langle \bar{\partial}_r\tilde{\alpha}(W_t), \widetilde{W}_t \right\rangle$ and $\left\langle -\bar{\partial}_r\mathcal{L}(W_t), \widetilde{W}_t \right\rangle$ are nonnegative, and thus

$$\left\langle \bar{\partial}_r\tilde{\alpha}(W_t), -\bar{\partial}_r\mathcal{L}(W_t) \right\rangle = \left\|\bar{\partial}_r\tilde{\alpha}(W_t)\right\|\left\|\bar{\partial}_r\mathcal{L}(W_t)\right\|.$$

Lemmas C.2 and C.3 also imply that $\bar{\partial}_\perp\tilde{\alpha}(W_t)$ and $-\bar{\partial}_\perp\mathcal{L}(W_t)$ point to the same direction, and thus

$$\left\langle \bar{\partial}_\perp\tilde{\alpha}(W_t), -\bar{\partial}_\perp\mathcal{L}(W_t) \right\rangle = \left\|\bar{\partial}_\perp\tilde{\alpha}(W_t)\right\|\left\|\bar{\partial}_\perp\mathcal{L}(W_t)\right\|.$$

Now consider $\widetilde{W}_t$ and $\zeta_t$. Since $W_t$ is an arc, and $\|W_t\| \geq \|W_0\| > 0$, it follows that $\widetilde{W}_t$ is also an arc. Moreover, for a.e. $t \geq 0$,

$$\frac{d\widetilde{W}_t}{dt} = \frac{1}{\|W_t\|}\frac{dW_t}{dt} - \frac{1}{\|W_t\|}\widetilde{W}_t\left\langle \frac{dW_t}{dt}, \widetilde{W}_t \right\rangle = \frac{-\bar{\partial}_\perp\mathcal{L}(W_t)}{\|W_t\|}.$$

Since $\widetilde{W}_t$ is an arc, $d\widetilde{W}_t/dt$ and $\left\|d\widetilde{W}_t/dt\right\|$ are both integrable, and by definition of the curve length,

$$\zeta_t = \int_0^t \left\|\frac{d\widetilde{W}_t}{dt}\right\| dt,$$

and for a.e. $t \geq 0$ we have

$$\frac{d\zeta_t}{dt} = \left\|\frac{d\widetilde{W}_t}{dt}\right\| = \frac{\left\|\bar{\partial}_\perp\mathcal{L}(W_t)\right\|}{\|W_t\|}.$$

$\square$

Finally we prove the core Lemma 3.3, which directly implies Theorem 3.1.

*Proof of Lemma 3.3.* Recall that $\tilde{\alpha}_t$ denotes $\tilde{\alpha}(W_t)$, and $a = \lim_{t\to\infty}\tilde{\alpha}_t$.

First note that if $\tilde{\alpha}_{t_0} = a$ for some finite $t_0$, then $d\tilde{\alpha}_t/dt = 0$ for a.e. $t \geq 0$. Lemma 3.4 then implies for a.e. $t \geq 0$ that $\left\|\bar{\partial}_\perp\mathcal{L}(W_t)\right\| = 0$ and $d\zeta_t/dt = 0$, and then Lemma 3.3 trivially holds. Below we assume $\tilde{\alpha}_t < a$ for all finite $t \geq 0$, and fix an arbitrary $\kappa \in (L/2, L)$. We consider two cases.

1. Lemma 3.5 implies that there exists $\nu_1 > 0$ and a definable desingularizing function $\Psi_1$ on $[0, \nu_1)$, such that if $W$ satisfies $\|W\| > 1$, and $\tilde{\alpha}(W) > a - \nu_1$, and

$$\left\|\bar{\partial}_\perp\tilde{\alpha}(W)\right\| \geq \frac{\tilde{\alpha}_0}{2\ln(n)+1}\|W\|^{L-\kappa}\left\|\bar{\partial}_r\tilde{\alpha}(W)\right\|, \tag{11}$$

   then

$$\Psi_1'\left(a - \tilde{\alpha}(W)\right)\|W\|\left\|\bar{\partial}\tilde{\alpha}(W)\right\| \geq 1. \tag{12}$$

   Now consider $t$ large enough such that $\|W_t\| > 1$, and $\tilde{\alpha}_t > a - \nu_1$, and $\tilde{\alpha}_0\|W_t\|^{L-\kappa}/(2\ln(n)+1) \geq 1$, and moreover assume eq. (11) holds for $W_t$. We have

$$\left\|\bar{\partial}_\perp\tilde{\alpha}(W_t)\right\| \geq \left\|\bar{\partial}_r\tilde{\alpha}(W_t)\right\|, \quad \text{and thus} \quad \left\|\bar{\partial}_\perp\tilde{\alpha}(W_t)\right\| \geq \frac{1}{2}\left\|\bar{\partial}\tilde{\alpha}(W_t)\right\|.$$

Therefore Lemma 3.4 implies

$$\frac{\mathrm{d}\tilde{\alpha}_t}{\mathrm{d}t} \geq \left\|\bar{\partial}_\perp \tilde{\alpha}(W_t)\right\|\left\|\bar{\partial}_\perp \mathcal{L}(W_t)\right\|$$

$$= \|W_t\|\left\|\bar{\partial}_\perp \tilde{\alpha}(W_t)\right\| \frac{\mathrm{d}\zeta_t}{\mathrm{d}t}$$

$$\geq \frac{1}{2}\|W_t\|\left\|\bar{\partial}\tilde{\alpha}(W_t)\right\| \frac{\mathrm{d}\zeta_t}{\mathrm{d}t}. \qquad (13)$$

Consequently, eqs. (12) and (13) imply that

$$\frac{\mathrm{d}\tilde{\alpha}_t}{\mathrm{d}t} \geq \frac{1}{2\Psi_1'\left(a - \tilde{\alpha}_t\right)} \frac{\mathrm{d}\zeta_t}{\mathrm{d}t}.$$

2. On the other hand, Lemma 3.6 implies that there exists $\nu_2 > 0$ and a definable desingulariz-ing function $\Psi_2$ on $[0, \nu_2)$, such that if $\|W\| > 1$, and $\tilde{\alpha}(W) > a - \nu_2$, then

$$\max\left\{1, \frac{2}{2\kappa - L}\right\} \Psi_2'\left(a - \tilde{\alpha}(W)\right) \|W\|^{2\kappa - L + 1}\left\|\bar{\partial}\tilde{\alpha}(W)\right\| \geq 1. \qquad (14)$$

Now consider $t$ large enough such that $\|W_t\| > 1$, and $\tilde{\alpha}_t > a - \nu_2$, and $\tilde{\alpha}_0\|W_t\|^{L-\kappa}/(2\ln(n) + 1) \geq 1$, and moreover

$$\left\|\bar{\partial}_\perp \tilde{\alpha}(W_t)\right\| \leq \frac{\tilde{\alpha}_0}{2\ln(n) + 1}\|W_t\|^{L-\kappa}\left\|\bar{\partial}_r \tilde{\alpha}(W_t)\right\|. \qquad (15)$$

Note that eq. (15) is the opposite to eq. (11). Lemmas C.2 and C.4 implies that

$$\left\|\bar{\partial}_r \alpha(W_t)\right\| = \frac{L\beta(W_t)}{\|W_t\|} \geq \frac{L\alpha(W_t)}{\|W_t\|} = L\tilde{\alpha}_t\|W_t\|^{L-1} \geq L\tilde{\alpha}_0\|W_t\|^{L-1}, \qquad (16)$$

while Lemma C.3 implies that

$$\left\|\bar{\partial}_r \tilde{\alpha}(W_t)\right\| = L\frac{\beta(W_t) - \alpha(W_t)}{\|W_t\|^{L+1}} \leq \frac{L(2\ln(n) + 1)}{\|W_t\|^{L+1}},$$

and thus

$$\left\|\bar{\partial}_r \alpha(W_t)\right\| \geq \frac{\tilde{\alpha}_0}{2\ln(n) + 1}\|W_t\|^{2L}\left\|\bar{\partial}_r \tilde{\alpha}(W_t)\right\|. \qquad (17)$$

On the other hand, $\bar{\partial}_\perp \alpha(W_t) = \|W_t\|^L \bar{\partial}_\perp \tilde{\alpha}(W_t)$ by Lemma C.3, which implies the follow-ing in light of eqs. (15) and (17):

$$\left\|\bar{\partial}_r \alpha(W_t)\right\| \geq \frac{\tilde{\alpha}_0}{2\ln(n) + 1}\|W_t\|^{2L}\left\|\bar{\partial}_r \tilde{\alpha}(W_t)\right\|$$

$$\geq \|W_t\|^{L+\kappa}\left\|\bar{\partial}_\perp \tilde{\alpha}(W_t)\right\|$$

$$= \|W_t\|^\kappa\left\|\bar{\partial}_\perp \alpha(W_t)\right\|.$$

By Lemma C.2, $\bar{\partial}\alpha(W_t)$ is parallel to $\bar{\partial}\mathcal{L}(W_t)$, therefore

$$\left\|\bar{\partial}_r \mathcal{L}(W_t)\right\| \geq \|W_t\|^\kappa\left\|\bar{\partial}_\perp \mathcal{L}(W_t)\right\|. \qquad (18)$$

Moreover, if $\tilde{\alpha}_0\|W_t\|^{L-\kappa}/(2\ln(n) + 1) \geq 1$, then the triangle inequality implies

$$\left\|\bar{\partial}\tilde{\alpha}(W_t)\right\| \leq \left\|\bar{\partial}_\perp \tilde{\alpha}(W_t)\right\| + \left\|\bar{\partial}_r \tilde{\alpha}(W_t)\right\| \leq \frac{2\tilde{\alpha}_0}{2\ln(n) + 1}\|W_t\|^{L-\kappa}\left\|\bar{\partial}_r \tilde{\alpha}(W_t)\right\|,$$

or

$$\left\|\bar{\partial}_r \tilde{\alpha}(W_t)\right\| \geq \frac{2\ln(n) + 1}{2\tilde{\alpha}_0}\|W_t\|^{\kappa - L}\left\|\bar{\partial}\tilde{\alpha}(W_t)\right\|. \qquad (19)$$

Now Lemma 3.4 and eqs. (18) and (19) imply

$$
\begin{aligned}
\frac{\mathrm{d}\tilde{\alpha}_t}{\mathrm{d}t} &\geq \left\|\bar{\partial}_r\tilde{\alpha}(W_t)\right\|\left\|\bar{\partial}_r\mathcal{L}(W_t)\right\| \\
&\geq \frac{2\ln(n)+1}{2\tilde{\alpha}_0}\|W_t\|^{2\kappa-L}\left\|\bar{\partial}\tilde{\alpha}(W_t)\right\|\left\|\bar{\partial}_\perp\mathcal{L}(W_t)\right\| \\
&= \frac{2\ln(n)+1}{2\tilde{\alpha}_0}\|W_t\|^{2\kappa-L+1}\left\|\bar{\partial}\tilde{\alpha}(W_t)\right\|\frac{\mathrm{d}\zeta_t}{\mathrm{d}t}.
\end{aligned}
$$

Then eq. (14) further implies

$$
\frac{\mathrm{d}\tilde{\alpha}_t}{\mathrm{d}t} \geq \frac{2\ln(n)+1}{2\tilde{\alpha}_0\max\{1,2/(2\kappa-L)\}}\frac{1}{\Psi_2'\left(a-\tilde{\alpha}_t\right)}\frac{\mathrm{d}\zeta_t}{\mathrm{d}t}.
$$

Since $\Psi_1'-\Psi_2'$ is definable, it is nonnegative or nonpositive on some interval $(0,\nu)$. Let $\Psi'=\max\{\Psi_1',\Psi_2'\}$ on $(0,\nu)$. Now for a.e. large enough $t$ such that $\|W_t\|>1$, and $\tilde{\alpha}_t>a-\nu$, and $\tilde{\alpha}_0\|W_t\|^{L-\kappa}/(2\ln(n)+1)\geq 1$, it holds that

$$
\frac{\mathrm{d}\tilde{\alpha}_t}{\mathrm{d}t} \geq \frac{1}{c\Psi'\left(a-\tilde{\alpha}_t\right)}\frac{\mathrm{d}\zeta_t}{\mathrm{d}t}
$$

for some constant $c>0$. Lemma 3.3 then follows. $\qquad\square$

# D  Omitted proofs from Section 4

We first give the following technical result.

**Lemma D.1.** *Suppose $f:\mathbb{R}^n\to\mathbb{R}$ is $L$-positively homogeneous for some $L>0$ and has a locally Lipschitz gradient at all nonzero $x\in\mathbb{R}^n$. Then $\nabla f$ is $(L-1)$-positively homogeneous: given any nonzero $x$ and $c>0$, it holds that*

$$
\nabla f(cx)=c^{L-1}\nabla f(x).
$$

*If $\nabla f$ is differentiable at a nonzero $x$, then for any $c>0$, it holds that*

$$
\nabla^2 f(cx)=c^{L-2}\nabla^2 f(x).
$$

*Moreover, there exists $K_\sigma>0$ such that for any $\|x\|=1$, if $\nabla^2 f(x)$ exists, then $\left\|\nabla^2 f(x)\right\|_\sigma\leq K_\sigma$.*

*Proof.* By definition,

$$
\lim_{\|y\|\downarrow 0}\frac{f(x+y)-f(x)-\left\langle\nabla f(x),y\right\rangle}{\|y\|}=0.
$$

On the other hand, by homogeneity,

$$
f(cx+z)-f(cx)-\left\langle c^{L-1}\nabla f(x),z\right\rangle=c^L\left(f\left(x+\frac{z}{c}\right)-f(x)-\left\langle\nabla f(x),\frac{z}{c}\right\rangle\right).
$$

Therefore

$$
\lim_{\|z\|\downarrow 0}\frac{f(cx+z)-f(cx)-\left\langle c^{L-1}\nabla f(x),z\right\rangle}{\|z\|}=c^{L-1}\lim_{\|z\|\downarrow 0}\frac{f\left(x+\frac{z}{c}\right)-f(x)-\left\langle\nabla f(x),\frac{z}{c}\right\rangle}{\|z/c\|}=0,
$$

which proves the claim. The homogeneity of $\nabla^2 f$ when it exists can be proved in the same way.

To get $K_\sigma$, note that for any $\|x\|=1$, there exists an open neighborhood $U_x$ of $x$ on which $\nabla f$ is $K_x$-Lipschitz continuous, and thus the spectral norm of $\nabla^2 f$ is bounded by $K_x$ when it exists. All the $U_x$ form an open cover of the compact unit sphere, and thus has a finite subcover, which implies the claim. $\qquad\square$

Below we estimate various quantities using Lemma D.1.

**Lemma D.2.** *Suppose $\ell \in \{\ell_{\exp}, \ell_{\log}\}$, all $p_i$ are L-positively homogeneous for some $L > 0$, and all $\nabla p_i$ are locally Lipschitz. For any $W$ such that $\mathcal{L}(W) < \ell(0)$, it holds that $\beta(W)/\|W\|^L$ and $\|\nabla\alpha(W)\|/\|W\|^{L-1}$ are bounded.*

*Proof.* Since $p_i(W)$ is continuous, it is bounded on the unit sphere. Because it is $L$-positively homogeneous, $p_i(W)/\|W\|^L$ is bounded on $\mathbb{R}^k$. Lemma C.4 implies that $\beta(W) - 2\ln(n) - 1 \leq \alpha(W) \leq \min_{1 \leq i \leq n} p_i(W)$, and it follows that $\beta(W)/\|W\|^L$ is bounded.

Recall that

$$\nabla\alpha(W) = \sum_{i=1}^n \frac{\partial\pi}{\partial p_i}\nabla p_i(W),$$

where $\pi$ is defined in eq. (10) and all partial derivatives are evaluated at $p(W) := (p_1(W), \ldots, p_n(W))$. It is shown in the proof of Lemma C.4 that $\|\pi(p)\|_1 \leq 2$. Moreover, Lemma D.1 implies that all $\|\nabla p_i(W)\|/\|W\|^{L-1}$ are bounded. Consequently, $\|\nabla\alpha(W)\|/\|W\|^{L-1}$ is bounded. $\square$

Recall the definition of $\mathcal{J}$:

$$\mathcal{J}(W) := \frac{\|\nabla\alpha(W)\|^2}{\|W\|^{2L-2}}.$$

If all $\nabla p_i$ are locally Lipschitz, then $\mathcal{J}$ is also locally Lipschitz. We further have the following result.

**Lemma D.3.** *Under the same conditions as Lemma D.2, for any $W$ satisfying $\mathcal{L}(W) < \ell(0)$ and any $W^* \in \partial\mathcal{J}(W)$,*

$$\langle W^*, -\nabla\mathcal{L}(W)\rangle \leq -K\ell'\left(\alpha(W)\right)\|W\|^{L-2}\sin(\theta)^2$$

*for some constant $K > 0$, where $\theta$ denotes the angle between $W$ and $-\nabla\mathcal{L}(W)$.*

*Proof.* Let $D'$ denote the set of $W$ where all $\nabla p_i$ are differentiable, and let $S_0$ denote the set of $W$ where $\mathcal{L}(W) < \ell(0)$. We only need to prove the lemma on $D' \cap S_0$, since for any $W \in S_0$ it follows from [Clarke, 1983, Theorem 2.5.1] that

$$\partial\mathcal{J}(W) = \text{conv}\left\{\lim \nabla\mathcal{J}(W_i)\big| W_i \to W, W_i \in D' \cap S_0\right\}.$$

Below we fix an arbitrary $W \in D' \cap S_0$. All the partial derivatives below with respect to $p_i$ are evaluated at $p(W) := (p_1(W), \ldots, p_n(W))$. Recall that

$$\nabla\alpha(W) = \sum_{i=1}^n \frac{\partial\pi}{\partial p_i}\nabla p_i(W),$$

where $\pi$ is defined in eq. (10). Since $\nabla p_i$ are also differentiable at $W$, we have

$$\nabla^2\alpha(W) = \sum_{i=1}^n\sum_{j=1}^n\left(\frac{\partial^2\pi}{\partial p_i\partial p_j}\nabla p_i(W)\nabla p_j(W)^\mathsf{T}\right) + \sum_{i=1}^n \frac{\partial\pi}{\partial p_i}\nabla^2 p_i(W). \tag{20}$$

Now for any $W \in D' \cap S_0$, we have (recall that $\widetilde{W} = W/\|W\|$)

$$\nabla\mathcal{J}(W) = \frac{2\nabla^2\alpha(W)\nabla\alpha(W)}{\|W\|^{2L-2}} - \frac{\|\nabla\alpha(W)\|^2}{\|W\|^{4L-4}} \cdot (2L-2)\|W\|^{2L-3}\widetilde{W}$$

$$= \frac{2\nabla^2\alpha(W)\nabla\alpha(W)}{\|W\|^{2L-2}} - \frac{(2L-2)\|\nabla\alpha(W)\|^2}{\|W\|^{2L}}W,$$

and thus

$$\frac{\|W\|^{2L}}{2}\frac{\langle\nabla\mathcal{J}(W),-\nabla\mathcal{L}(W)\rangle}{-\ell'\left(\alpha(W)\right)}$$

$$=\frac{\|W\|^{2L}}{2}\langle\nabla\mathcal{J}(W),\nabla\alpha(W)\rangle$$

$$=\|W\|^2\nabla\alpha(W)^\mathsf{T}\nabla^2\alpha(W)\nabla\alpha(W)-(L-1)\|\nabla\alpha(W)\|^2\langle W,\nabla\alpha(W)\rangle. \qquad (21)$$

Comparing eqs. (20) and (21), first note that

$$\sum_{i=1}^n\sum_{j=1}^n\frac{\partial^2\pi}{\partial p_i\partial p_j}\nabla\alpha(W)^\mathsf{T}\nabla p_i(W)\nabla p_j(W)^\mathsf{T}\nabla\alpha(W)\le 0,$$

since $\pi$ is concave by Lemma C.5, and moreover

$$\langle W,\nabla\alpha(W)\rangle=\sum_{i=1}^n\frac{\partial\pi}{\partial p_i}\langle W,\nabla p_i(W)\rangle=L\sum_{i=1}^n\frac{\partial\pi}{\partial p_i}p_i(W).$$

Therefore eq. (21) is upper bounded by

$$\|W\|^2\sum_{i=1}^n\frac{\partial\pi}{\partial p_i}\nabla\alpha(W)^\mathsf{T}\nabla^2 p_i(W)\nabla\alpha(W)-L(L-1)\|\nabla\alpha(W)\|^2\sum_{i=1}^n\frac{\partial\pi}{\partial p_i}p_i(W). \qquad (22)$$

Let $\nabla_r\alpha(W)$ and $\nabla_\perp\alpha(W)$ denote the radial and spherical part of $\nabla\alpha(W)$, respectively. Let $\theta$ denote the angle between $W$ and $\nabla\alpha(W)$. Lemmas C.2 and C.4 imply that

$$\langle W,\nabla\alpha(W)\rangle=L\beta(W)>0,$$

and thus $\theta$ is between $0$ and $\pi/2$. Now Lemma D.1 and the proof of Lemma C.1 imply that

$$\|W\|^2\nabla_r\alpha(W)^\mathsf{T}\nabla^2 p_i(W)\nabla_r\alpha(W)=\cos(\theta)^2\|\nabla\alpha(W_t)\|^2 W^\mathsf{T}\nabla^2 p_i(W)W$$

$$=\cos(\theta)^2\|\nabla\alpha(W_t)\|^2\cdot L(L-1)p_i(W)$$

$$\le\|\nabla\alpha(W_t)\|^2\cdot L(L-1)p_i(W). \qquad (23)$$

Moreover,

$$2\|W\|^2\nabla_\perp\alpha(W)^\mathsf{T}\nabla^2 p_i(W)\nabla_r\alpha(W)=2\|W\|\|\nabla\alpha(W)\|\cos(\theta)\left\langle\nabla_\perp\alpha(W),\nabla^2 p_i(W)W\right\rangle$$

$$=2(L-1)\|W\|\|\nabla\alpha(W)\|\cos(\theta)\left\langle\nabla_\perp\alpha(W),\nabla p_i(W)\right\rangle,$$

and thus by Lemma C.2,

$$2\|W\|^2\sum_{i=1}^n\frac{\partial\pi}{\partial p_i}\nabla_\perp\alpha(W)^\mathsf{T}\nabla^2 p_i(W)\nabla_r\alpha(W)$$

$$=2(L-1)\|W\|\|\nabla\alpha(W)\|\cos(\theta)\left\langle\nabla_\perp\alpha(W),\nabla\alpha(W)\right\rangle$$

$$=2(L-1)\|W\|\|\nabla\alpha(W)\|^3\cos(\theta)\sin(\theta)^2$$

$$=2L(L-1)\|\nabla\alpha(W)\|^2\sin(\theta)^2\beta(W). \qquad (24)$$

In addition, the proof of Lemma C.4 shows that $\|\pi(p)\|_1\le 2$, and Lemma D.1 ensures that $\|\nabla^2 f\|_\sigma$ has a uniform bound $K_\sigma$ on the unit sphere, therefore

$$\|W\|^2\sum_{i=1}^n\frac{\partial\pi}{\partial p_i}\nabla_\perp\alpha(W)^\mathsf{T}\nabla^2 p_i(W)\nabla_\perp\alpha(W)\le 2\|W\|^2\|\nabla\alpha(W)\|^2\sin(\theta)^2\cdot K_\sigma\|W\|^{L-2}$$

$$=2K_\sigma\|W\|^L\|\nabla\alpha(W)\|^2\sin(\theta)^2. \qquad (25)$$

Combining eqs. (21) to (25) gives

$$\frac{\langle\nabla\mathcal{J}(W),-\nabla\mathcal{L}(W)\rangle}{-\ell'\left(\alpha(W)\right)}\le\frac{4\left(K_\sigma\|W\|^L+L(L-1)\beta(W)\right)\|\nabla\alpha(W)\|^2}{\|W\|^{2L}}\sin(\theta)^2.$$

Invoking Lemma D.2 then gives

$$\langle\nabla\mathcal{J}(W),-\nabla\mathcal{L}(W)\rangle\le -K\ell'\left(\alpha(W)\right)\|W\|^{L-2}\sin(\theta)^2$$

for some constant $K>0$. $\qquad\square$

The following result helps us control $\theta_t$.

**Lemma D.4.** *Under the same condition as Lemma D.2 and Assumption 2.2, it holds that*

$$\int_0^\infty -\ell'\left(\alpha(W_t)\right)\|W_t\|^{L-2}\tan(\theta_t)^2\,\mathrm{d}t < \infty.$$

*Proof.* Recall that $\tilde{\alpha}_t = \alpha(W_t)/\|W_t\|^L$ is nondecreasing with a limit $a$, and thus $\mathrm{d}\tilde{\alpha}_t/\mathrm{d}t$ is integrable. Now Lemmas 3.4, C.2 and C.3 imply that

$$\frac{\mathrm{d}\tilde{\alpha}_t}{\mathrm{d}t} \geq \|\nabla_\perp\tilde{\alpha}(W_t)\|\|\nabla_\perp\mathcal{L}(W_t)\| = \frac{\|\nabla_\perp\alpha(W_t)\|\|\nabla_\perp\mathcal{L}(W_t)\|}{\|W_t\|^L} = \frac{-\ell'\left(\alpha(W_t)\right)\|\nabla_\perp\alpha(W_t)\|^2}{\|W_t\|^L},$$

and moreover

$$\|\nabla_\perp\alpha(W_t)\| = \|\nabla_r\alpha(W_t)\|\tan(\theta_t) = \frac{L\beta(W_t)}{\|W_t\|}\tan(\theta_t).$$

Therefore

$$\frac{\mathrm{d}\tilde{\alpha}_t}{\mathrm{d}t} \geq -\ell'\left(\alpha(W_t)\right)\cdot L^2\tan(\theta_t)^2\frac{\beta(W_t)^2}{\|W_t\|^{L+2}}.$$

Since $\beta(W_t)/\|W_t\|^L$ is bounded due to Lemma D.2, the proof is finished. $\qquad\square$

Now we can prove Theorem 4.1.

*Proof of Theorem 4.1.* Fix an arbitrary $\epsilon \in (0,1)$, and let $J_t$ denote $J(W_t)$. Recall that $\lim_{t\to\infty}\alpha(W_t)/\|W_t\|^L = a$. Lemma C.4 then implies $\lim_{t\to\infty}\beta(W_t)/\|W_t\|^L = a$, and thus we can find $t_1$ such that for any $t > t_1$,

$$a\left(1-\frac{\epsilon}{6}\right) < \frac{\beta(W_t)}{\|W_t\|^L} = \frac{1}{L}\left\langle\frac{\nabla\alpha(W_t)}{\|W_t\|^{L-1}},\frac{W_t}{\|W_t\|_F}\right\rangle < a\left(1+\frac{\epsilon}{6}\right). \tag{26}$$

Moreover, Lemmas D.3, D.4 and B.9 imply that there exists $t_2$ such that for any $t' > t > t_2$,

$$J_{t'} - J_t < \left(\frac{aL\epsilon}{6}\right)^2. \tag{27}$$

[Lyu and Li, 2019, Corollary C.10] implies that there exists $t_3 > \max\{t_1, t_2\}$ such that

$$\frac{1}{\cos(\theta_{t_2})^2} - 1 < \frac{\epsilon}{3}, \quad\text{and thus}\quad \frac{1}{\cos(\theta_{t_2})} < 1 + \frac{\epsilon}{6}. \tag{28}$$

We claim that $\delta_t < 1 + \epsilon$ for any $t > t_3$.

To see this, note that eqs. (26) and (28) imply

$$\sqrt{J_{t_2}} = \frac{\|\nabla\alpha(W_{t_2})\|}{\|W_{t_2}\|^{L-1}} < aL\left(1+\frac{\epsilon}{6}\right)\frac{1}{\cos(\theta_{t_2})} < aL\left(1+\frac{\epsilon}{6}\right)^2 < aL\left(1+\frac{\epsilon}{2}\right).$$

Moreover, using eq. (27), for any $t > t_2$,

$$\sqrt{J_t} = \sqrt{J_{t_2} + J_t - J_{t_2}} < \sqrt{J_{t_2} + \left(\frac{\gamma L\epsilon}{6}\right)^2} < \sqrt{J_{t_2}} + \frac{aL\epsilon}{6} < aL\left(1+\frac{2\epsilon}{3}\right),$$

and thus

$$\frac{1}{\cos(\theta_t)} = \frac{\sqrt{J_t}}{L\beta(W_t)/\|W_t\|^L} < \frac{aL\left(1+2\epsilon/3\right)}{aL(1-\epsilon/6)} < 1+\epsilon.$$

Since $\epsilon$ is arbitrary, we have $\lim_{t\to\infty}\theta_t = 0$.

If all $p_i$ are $C^2$, then the above proof holds without definability: it is only used in eq. (27) to ensure the chain rule, which always holds for $C^2$ functions. $\qquad\square$

# E  Global margin maximization proofs for Section 4.2

This section often works with subscripted subsets of parameters, for instance per-layer matrices $(A_1(t), \ldots, A_L(t))$, or per-node weights $(w_1(t), \ldots, w_m(t))$; to declutter slightly, we will drop "$(t)$" throughout when it is otherwise clear.

First, a technical lemma regarding directional convergence and alignment properties inherited by these subsets of $W_t$. This will be used in both the deep linear case and in the 2-homogeneous case.

**Lemma E.1.** *Suppose the conditions for Theorems 3.1 and 4.1 hold. Let $(U_1(t), \ldots, U_r(t))$ be any partition of $W_t$, and set $s_j(t) := \|U_j(t)\|^L / \|W_t\|^L$. Then $s(t)$ converges to some $\bar{s}$, and for each $j$,*

$$\lim_{t \to \infty} \frac{\|U_j\| \cdot \|\nabla_{U_j} \mathcal{L}(W)\|}{\|W\| \cdot \|\nabla_W \mathcal{L}(W)\|} = \lim_{t \to \infty} \frac{\langle U_j, -\nabla_{U_j} \mathcal{L}(W) \rangle}{\|W\| \cdot \|\nabla_W \mathcal{L}(W)\|},$$

*and moreover $\bar{s}_j > 0$ implies*

$$\lim_{t \to \infty} \frac{\|U_j\|}{\|W\|} = \lim_{t \to \infty} \frac{\|\nabla_{U_j} \mathcal{L}(W)\|}{\|\nabla_W \mathcal{L}(W)\|} = \lim_{t \to \infty} \frac{\|\nabla_{U_j} \alpha(W)\|}{\|\nabla_W \alpha(W)\|} = \bar{s}_j^{1/L},$$

*and*

$$\lim_{t \to \infty} \frac{\langle U_j, -\nabla_{U_j} \mathcal{L}(W) \rangle}{\|U_j\| \cdot \|\nabla_{U_j} \mathcal{L}(W)\|} = \lim_{t \to \infty} \frac{\langle U_j, \nabla_{U_j} \alpha(W) \rangle}{\|U_j\| \cdot \|\nabla_{U_j} \alpha(W)\|} = 1,$$

*and*

$$\lim_{t \to \infty} \frac{\langle U_j, \nabla_{U_j} \alpha(W) \rangle}{\|U_j\|^L} = \lim_{t \to \infty} \frac{\|\nabla_{U_j} \alpha(W)\|}{\|U_j\|^{L-1}} = a\bar{s}^{(2-L)/L} L.$$

*Proof.* First note that $s(t)$ converges since $W_t / \|W_t\|$ converges, and alignment grants

$$\bar{s}_j^{1/L} = \lim_{t \to \infty} \frac{\|U_j\|}{\|W\|} = \lim_{t \to \infty} \frac{\|\nabla_{U_j} \mathcal{L}(W)\|}{\|\nabla_W \mathcal{L}(W)\|}. \tag{29}$$

By directional convergence (cf. Theorem 3.1), alignment (cf. Theorem 4.1), and Cauchy-Schwarz,

$$
\begin{aligned}
-1 &= \lim_{t \to \infty} \frac{\langle W, \nabla_W \mathcal{L}(W) \rangle}{\|W\| \cdot \|\nabla_W \mathcal{L}(W)\|} \\
&= \lim_{t \to \infty} \frac{\sum_j \langle U_j, \nabla_{U_j} \mathcal{L}(W) \rangle}{\|W\| \cdot \|\nabla_W \mathcal{L}(W)\|} \\
&\geq -\lim_{t \to \infty} \frac{\sum_j \|U_j\| \cdot \|\nabla_{U_j} \mathcal{L}(W)\|}{\|W\| \cdot \|\nabla_W \mathcal{L}(W)\|} \\
&\geq -\lim_{t \to \infty} \frac{\sqrt{\sum_j \|U_j\|^2} \cdot \sqrt{\sum_j \|\nabla_{U_j} \mathcal{L}(W)\|^2}}{\|W\| \cdot \|\nabla_W \mathcal{L}(W)\|} = -1,
\end{aligned}
$$

which starts and ends with $-1$ and is thus a chain of equalities. Applying eq. (29) and he equality case of Cauchy-Schwarz to each $j$ with $\bar{s}_j > 0$,

$$
\begin{aligned}
\bar{s}_j^{2/L} &= \lim_{t \to \infty} \frac{\|U_j\| \cdot \|\nabla_{U_j} \mathcal{L}(W)\|}{\|W\| \cdot \|\nabla_W \mathcal{L}(W)\|} = \lim_{t \to \infty} \frac{\langle U_j, -\nabla_{U_j} \mathcal{L}(W) \rangle}{\|W\| \cdot \|\nabla_W \mathcal{L}(W)\|} \\
&= \lim_{t \to \infty} \frac{\langle U_j, -\nabla_{U_j} \mathcal{L}(W) \rangle}{\|U_j\| \cdot \|\nabla_{U_j} \mathcal{L}(W)\|} \left( \frac{\|U_j\| \cdot \|\nabla_{U_j} \mathcal{L}(W)\|}{\|W\| \cdot \|\nabla_W \mathcal{L}(W)\|} \right) \\
&= \bar{s}_j^{2/L} \lim_{t \to \infty} \frac{\langle U_j, -\nabla_{U_j} \mathcal{L}(W) \rangle}{\|U_j\| \cdot \|\nabla_{U_j} \mathcal{L}(W)\|},
\end{aligned}
$$

and thus

$$\lim_{t \to \infty} \frac{\langle U_j, -\nabla_{U_j} \mathcal{L}(W) \rangle}{\|U_j\| \cdot \|\nabla_{U_j} \mathcal{L}(W)\|} = 1.$$

The preceding statements used $\mathcal{L}(W)$; to obtain the analogous statements with $\alpha(W)$, note since $\ell' < 0$ that

$$\frac{\nabla_{U_j}\alpha(W)}{\|\nabla_{U_j}\alpha(W)\|} = \frac{\nabla_{U_j}\mathcal{L}(W)/\ell'(\alpha(W))}{\|\nabla_{U_j}\mathcal{L}(W)/\ell'(\alpha(W))\|} = \frac{-\nabla_{U_j}\mathcal{L}(W)}{\|\nabla_{U_j}\mathcal{L}(W)\|}.$$

For the final claim, note Theorem 4.1 and eq. (3) imply that

$$\lim_{t\to\infty} \frac{\|\nabla\alpha(W_t)\|}{\|W_t\|^{L-1}} = \lim_{t\to\infty} \frac{\langle\nabla\alpha(W_t), W_t\rangle}{\|W_t\|^L} = aL > 0,$$

and when $\bar{s}_j > 0$,

$$\lim_{t\to\infty} \frac{\langle U_j, \nabla_{U_j}\alpha(W_t)\rangle}{\|U_j\|^L} = \lim_{t\to\infty} \frac{\|U_j\| \cdot \|\nabla_{U_j}\alpha(W)\|}{\|U_j\|^L} = \lim_{t\to\infty} \frac{\|\nabla_{U_j}\alpha(W_t)\|}{\|U_j\|^{L-1}}$$

$$= \lim_{t\to\infty} \frac{\bar{s}^{1/L}\|\nabla_W\alpha(W)\|}{\bar{s}^{(L-1)/L}\|W\|^{L-1}} = aL\bar{s}^{(2-L)/L}.$$

$\square$

Applying the preceding lemma to network layers, we handle the deep linear case as follows.

*Proof of Proposition 4.2.* For convenience, write $A_j$ instead of $A_j(t)$ when time $t$ is clear, and also $u := A_j \cdots A_1$ and $\nabla_u\mathcal{L}(W) = \sum_i \ell'(y_i u^\mathsf{T} x_i)y_i x_i$. By this notation,

$$\nabla_{A_j}\mathcal{L}(W) = \sum_i \ell'(y_i u^\mathsf{T} x_i)y_i(A_L \cdots A_{j+1})^\mathsf{T}(A_{j-1}\cdots A_1 x_i)^\mathsf{T}$$

$$= (A_L \cdots A_{j+1})^\mathsf{T}(A_{j-1}\cdots A_1\nabla_u\mathcal{L}(W))^\mathsf{T},$$

where $(A_L \cdots A_{j+1})^\mathsf{T}$ is a column vector, and $(A_{j-1}\cdots A_1\nabla_u\mathcal{L}(W))^\mathsf{T}$ is a row vector, and moreover $\langle A_j, \nabla_{A_j}\mathcal{L}(W)\rangle = \langle u, \nabla_u\mathcal{L}(W)\rangle$, where this last inner product does not depend on $j$.

Applying the subset-alignment of Lemma E.1 to layers $(A_j, \ldots, A_1)$ gives, for each $j$,

$$\bar{s}_j^{2/L} = \lim_{t\to\infty} \frac{\|A_j\| \cdot \|\nabla_{A_j}\mathcal{L}(W)\|}{\|W\| \cdot \|\nabla_W\mathcal{L}(W)\|} = \lim_{t\to\infty} \frac{\langle A_j, -\nabla_{A_j}\mathcal{L}(W)\rangle}{\|W\| \cdot \|\nabla_W\mathcal{L}(W)\|} = \lim_{t\to\infty} \frac{-\langle u, \nabla_u\mathcal{L}(W)\rangle}{\|W\| \cdot \|\nabla_W\mathcal{L}(W)\|},$$

whereby $\bar{s}_j$ is independent of $j$, which can only mean $\bar{s}_j^{2/L} = 1/L > 0$ for all $j$, but more importantly $\|A_j(t)\| \to \infty$ for all $j$. By Lemma E.1, this means all layers align with their gradients.

Next it is proved by induction from $A_L$ to $A_1$ that there exist unit vectors $v_0, \ldots, v_L$ with $v_L = 1$ and $A_j/\|A_j\| = v_j v_{j-1}^\mathsf{T}$. The base case $A_L$ holds immediately, since $A_L$ is a row vector, meaning we can choose $v_L := 1$ and $v_{L-1} := A_L^\mathsf{T}/\|A_L\|$ since $A_L$ converges in direction. For the inductive step $A_j$ with $j < L$, note

$$\lim_{t\to\infty} \frac{\nabla_{A_j}\mathcal{L}(W)}{\|\nabla_{A_j}\mathcal{L}(W)\|} = \lim_{t\to\infty} \frac{(A_L \cdots A_{j+1})^\mathsf{T}(A_{j-1}\cdots A_1\nabla_u\mathcal{L}(W))^\mathsf{T}}{\|(A_L \cdots A_{j+1})^\mathsf{T}(A_{j-1}\cdots A_1\nabla_u\mathcal{L}(W))^\mathsf{T}\|}$$

$$= \lim_{t\to\infty} \frac{(A_L \cdots A_{j+1})^\mathsf{T}(A_{j-1}\cdots A_1\nabla_u\mathcal{L}(W))^\mathsf{T}}{\|(A_L \cdots A_{j+1})\|\|(A_{j-1}\cdots A_1\nabla_u\mathcal{L}(W))\|}$$

$$= \lim_{t\to\infty} \frac{(v_L v_{L-1}^\mathsf{T} \cdots v_{j+1}v_j^\mathsf{T})^\mathsf{T}(A_{j-1}\cdots A_1\nabla_u\mathcal{L}(W))^\mathsf{T}}{\|A_{j-1}\cdots A_1\nabla_u\mathcal{L}(W)\|}$$

$$= \lim_{t\to\infty} \frac{v_j(A_{j-1}\cdots A_1\nabla_u\mathcal{L}(W))^\mathsf{T}}{\|A_{j-1}\cdots A_1\nabla_u\mathcal{L}(W)\|}.$$

Since $v_j$ is a fixed unit vector and since $\nabla_{A_j}\mathcal{L}(W)$ converges in direction, the row vector part of the above expression must also converge to some fixed unit vector $v_{j-1}^\mathsf{T}$, namely

$$\lim_{t\to\infty} \frac{\nabla_{A_j}\mathcal{L}(W)}{\|\nabla_{A_j}\mathcal{L}(W)\|} = -v_j v_{j-1}^\mathsf{T} \qquad \text{where } v_{j-1} := -\lim_{t\to\infty} \frac{A_{j-1}\cdots A_1\nabla_u\mathcal{L}(W)}{\|A_{j-1}\cdots A_1\nabla_u\mathcal{L}(W)\|}.$$

Since $A_j$ and $-\nabla_{A_j}\mathcal{L}(W)$ asymptotically align as above, then ${A_j}/{\|A_j\|} \to v_j v_{j-1}^\mathsf{T}$.

Now consider $v_0$ and $u$, where it still needs to be shown that $v_0 = u/\|u\|$. To this end, note

$$1 \geq \lim_{t\to\infty} \frac{v_0^\mathsf{T} u}{\|u\|} = \lim_{t\to\infty} \frac{v_0^\mathsf{T} A_L \cdots A_1}{\|A_L \cdots A_1\|} \geq \lim_{t\to\infty} \left( \frac{\|A_L\| \cdots \|A_1\|}{\|A_L\|_\sigma \cdots \|A_2\|_\sigma \|A_1\|} \right) v_0^\mathsf{T} \left( v_L v_{L-1}^\mathsf{T} \cdots v_1 v_0^\mathsf{T} \right)$$
$$= v_0^\mathsf{T} v_0 = 1,$$

whereby $u/\|u\| = v_0$. By a similar calculation,

$$-1 = \lim_{t\to\infty} \frac{\langle A_1, \nabla_{A_1}\mathcal{L}(W) \rangle}{\|A_1\| \cdot \|\nabla_{A_1}\mathcal{L}(W)\|} = \lim_{t\to\infty} \frac{\langle u, \nabla_u \mathcal{L}(W) \rangle}{\|A_1\| \cdot \|A_L \cdots A_2 \nabla_u \mathcal{L}(W)\|} = \lim_{t\to\infty} \frac{\langle u, \nabla_u \mathcal{L}(W) \rangle}{\|u\| \cdot \|\nabla_u \mathcal{L}(W)\|},$$

which means $u/\|u\|$ asymptotically satisfies the optimality conditions for the optimization problem

$$\min_{\|w\|\leq 1} \frac{1}{\|A_L \cdots A_1\|} \sum_i \ell\left( \|A_L \cdots A_1\| y_i x_i^\mathsf{T} w \right),$$

which is asymptotically solved by the unique maximum margin vector $\bar{u}$, which is guaranteed to exist since the data is linearly separable thanks to $\mathcal{L}(W_0) < \ell(0)$. $\qquad\square$

Before moving on to the 2-homogeneous case, we first produce another technical lemma, which we will use to control *dual variables* $q_i(t) := \partial\alpha/\partial p_i(W_t)$, which also appear in Proposition 4.3.

**Lemma E.2.** *Every accumulation point $\bar{q}$ of $\{q(t) \mid t \in \mathbb{N}\}$ satisfies $\bar{q} \in \Delta_n$ and*

$$\sum_i \bar{q}_i \left\langle \frac{W}{\|W\|}, \frac{\nabla_W p_i(W)}{\|W\|^{L-1}} \right\rangle = \lim_{t\to\infty} \left\langle \frac{W}{\|W\|}, \frac{\nabla_W \alpha(W)}{L\|W\|^{L-1}} \right\rangle = \min_i \lim_{t\to\infty} \frac{p_i(W_t)}{\|W_t\|^L} = a.$$

*Proof.* By Lemmas C.2 and C.4,

$$\lim_{t\to\infty} \frac{\alpha(W_t)}{\|W_t\|^L} = \lim_{t\to\infty} \left\langle \frac{W_t}{\|W_t\|}, \frac{\nabla_W \alpha(W_t)}{L\|W_t\|^{L-1}} \right\rangle = \lim_{t\to\infty} \min_i \frac{p_i(W_t)}{\|W_t\|^L} = a = \min_i \lim_{t\to\infty} \frac{p_i(W_t)}{\|W_t\|^L} = a.$$

Moreover, since $\lim_{z\to\infty} \frac{\ell_{\log}(z)}{\ell_{\exp}(z)} = 1$ and since $a > 0$ and $\|W_t\| \to \infty$, then $q(t)$ is asymptotically within the simplex, meaning $\lim_{t\to\infty} \min_{q'\in\Delta_n} \|q(t) - q'\| = 0$. Consequently, every accumulation point $\bar{q}$ of $\{q(t) : t \in \mathbb{N}\}$ satisfies $\bar{q} \in \Delta_n$, and

$$\sum_i \bar{q}_i \lim_{t\to\infty} \left\langle \frac{W}{\|W\|}, \frac{\nabla_W p_i(W)}{\|W\|^{L-1}} \right\rangle = \lim_{t\to\infty} \left\langle \frac{W}{\|W\|}, \frac{\nabla_W \alpha_i(W)}{L\|W\|^{L-1}} \right\rangle = \lim_{t\to\infty} \min_i \frac{p_i(W_t)}{\|W_t\|^L} = a.$$

$\qquad\square$

With this in hand, we can handle the 2-homogeneous case.

*Proof of Proposition 4.3.* Applying Lemma E.1 to the per-node weights $(w_1, \ldots, w_m)$, a limit $\bar{s}$ exists and due to 2-homogeneity satisfies $\bar{s} \in \Delta_m$. Whenever, $\bar{s}_j > 0$, then

$$\lim_{t\to\infty} 2 \sum_i q_i(t)\varphi_{ij}(\theta_j(t)) = \lim_{t\to\infty} \left\langle \theta_j(t), \sum_i q_i(t)\nabla_\theta \varphi_{ij}(\theta_j(t)) \right\rangle$$
$$= \lim_{t\to\infty} \left\langle \frac{w_j(t)}{\|w_j(t)\|}, \frac{\nabla_{w_j}\alpha(W_t)}{\|w_j(t)\|} \right\rangle = 2a\bar{s}^{0/2} = 2a.$$

Consequently, this means that either $\bar{s}_j > 0$ and $\lim_{t\to\infty} \sum_i q_i(t)\varphi_{ij}(\theta_j(t)) = a$, or else $\bar{s}_j = 0$ and by the choice $\bar{\theta}_j = 0$ then $\lim_{t\to\infty} \sum_i q_i(t)\varphi_{ij}(\theta_j(t)) = 0$. In particular, this means $\bar{s}_j > 0$ iff $\bar{\theta}_j$ attains the maximal value $a$, meaning $\bar{s}$ satisfies the Sion primal optimality conditions for the saddle point problem over the fixed points $(\bar{\theta}_1, \ldots, \bar{\theta}_m)$ [Chizat and Bach, 2020, Proposition D.3].

Now consider the dual variables $q_i(t) = \partial\alpha/\partial p_i(W_t)$. By Lemma E.2, any accumulation point $\bar{q}$ is an element of $\Delta_n$ and moreover is supported on those examples $i$ minimizing $p_i(\overline{W})$, which means $\bar{q}$

satisfies the Sion dual optimality conditions for the margin saddle point problem again over fixed points $(\bar\theta_1, \ldots, \bar\theta_m)$ [Chizat and Bach, 2020, Proposition D.3]. Thus applying the Sion Theorem over discrete domain $(\bar\theta_1, \ldots, \bar\theta_m)$ to the primal-dual optimal pair $(\bar s, \bar q)$ gives

$$\sum_i \bar q_i \sum_j \bar s_j \varphi_{ij}(\bar\theta_j) = \min_{q \in \Delta_n} \max_{s \in \Delta_m} \sum_i q_i \sum_j s_j \varphi_{ij}(\bar\theta_j) = \min_i \max_{s \in \Delta_m} \sum_j s_j \varphi_{ij}(\bar\theta_j),$$

and directional convergence of $\widetilde{W}_t$ combined with definition of $\bar q$ gives

$$\lim_{t \to \infty} \sum_i q_i(t) s_j(t) \varphi_{ij}(\theta_j(t)) = \sum_i \bar q_i \sum_j \bar s_j \varphi_{ij}(\bar\theta_j(t)).$$

Since $\bar q$ was an arbitrary accumulation point, it holds in general that

$$\lim_{t \to \infty} \sum_i q_i(t) s_j(t) \varphi_{ij}(\theta_j(t)) = \min_{q \in \Delta_n} \max_{s \in \Delta_m} \sum_i q_i \sum_j s_j \varphi_{ij}(\bar\theta_j).$$

Now for the global guarantee. Fix $t_0$ for now, and consider $(\theta_j)_{j=1}^m = (\theta_j(t_0))_{j=1}^m$ and their cover guarantee. For any signed measure $\nu$ on $\mathbb{S}^{d-1}$, we can partition $\mathbb{S}^{d-1}$ twice so that $(\nu(\theta_1), \nu(\theta_3), \ldots)$ partitions the negative mass of $\nu$ by associating it with the closest element amongst $(\theta_1, \theta_3, \ldots)$, all of which have negative coefficient in $\varphi_{ij}$, and also the positive mass of $\nu$ into $(\nu(\theta_2), \nu(\theta_4), \ldots)$; in this way, we now have converted $\nu$ on $\mathbb{S}^{d-1}$ into a discrete measure on $(\theta_1, \ldots, \theta_m)$. Noting that $z \mapsto \max\{0, z\}^2$ is 2-Lipschitz over $[-1, 1]$, and therefore for any $i$ and any unit norm $\theta, \theta'$ that

$$|\varphi_{ij}(\theta) - \varphi_{ij}(\theta')| = \left|\max\{0, x_i^\mathsf{T}\theta\}^2 - \max\{0, x_i^\mathsf{T}\theta'\}^2\right| \le 2\left|x_i^\mathsf{T}\theta - x_i^\mathsf{T}\theta'\right| \le 2\|\theta - \theta'\|,$$

then, letting "$\theta \to \theta_j$" denote the subset of $\mathbb{S}^{d-1}$ associated with $\theta_j$ as above (positively or negatively), and letting $\varphi_i(\theta) := y_i \max\{0, x_i^\mathsf{T}\theta\}^2$, for any $q$,

$$\left|\sum_i q_i \int \varphi_i(\theta)\, \mathrm{d}\nu(\theta) - \sum_i q_i \sum_j \nu(\theta_j) \varphi_{ij}(\theta_j)\right|$$

$$= \left|\sum_i q_i \sum_j \int_{\theta \to \theta_j} \varphi_i(\theta)\, \mathrm{d}\nu(\theta) - \sum_i q_i \sum_j \nu(\theta_j) \varphi_{ij}(\theta_j)\right|$$

$$\le \sum_i q_i \int_{\theta \to \theta_j} \sum_j |\varphi_{ij}(\theta) - \varphi_{ij}(\theta_j)|\, \mathrm{d}|\nu|(\theta)$$

$$\le 2 \sum_i q_i \int_{\theta \to \theta_j} \sum_j \|\theta - \theta_j\|\, \mathrm{d}|\nu|(\theta) \le 2\epsilon.$$

Thus

$$\min_{q \in \Delta_n} \max_{p \in \Delta_m} \sum_i q_i \sum_j p_j \varphi_{ij}(\theta_j) \le \min_{q \in \Delta_n} \max_{\nu \in \mathcal{P}(\mathbb{S}^{d-1})} \sum_i q_i \int \varphi_i(\theta)\, \mathrm{d}\nu(\theta)$$

$$\le 2\epsilon + \min_{q \in \Delta_n} \max_{p \in \Delta_m} \sum_i q_i \sum_j p_j \varphi_{ij}(\theta_j).$$

Next, for any $q \in \Delta_n$ and $s \in \Delta_m$, using the first part of the cover condition,

$$\sum_{i,j} q_i s_j (\varphi_{ij}(\bar\theta_j) - \varphi_{ij}(\theta_j(t_0)) \le \sum_{i,j} q_i s_j |\varphi_{ij}(\bar\theta_j) - \varphi_{ij}(\theta_j(t_0))| \le 2 \sum_{i,j} q_i s_j \|\bar\theta_j - \theta_j(t_0)\| \le 2\epsilon,$$

thus

$$\lim_{t\to\infty} \sum_{i,j} q_i s_j \varphi_{ij}(\theta_j) = \min_{q\in\Delta_n} \max_{s\in\Delta_m} \sum_{i,j} q_i s_j \varphi_{ij}(\bar{\theta}_j)$$

$$= \min_{q\in\Delta_n} \max_{s\in\Delta_m} \left[ \sum_{i,j} q_i s_j \varphi_{ij}(\theta_j(t_0)) - \sum_{i,j} q_i s_j \left( \varphi_{ij}(\theta_j(t_0)) - \varphi_{ij}(\bar{\theta}_j) \right) \right]$$

$$\geq \min_{q\in\Delta_n} \max_{s\in\Delta_m} \sum_{i,j} q_i s_j \varphi_i(\theta_j(t_0)) - 2\epsilon$$

$$\geq \min_{q\in\Delta_n} \max_{\nu\in\mathcal{P}(\mathbb{S}^{d-1})} \sum_i q_i \int \varphi_i(\theta) \, \mathrm{d}\nu(\theta) - 4\epsilon.$$

$\square$