[Reviews · NeurIPS 2020]

Review 1

Summary and Contributions: The paper considers the setting of Gradient Flow (GF) over homogeneous models (a broad class of models including neural networks with fully-connected and convolutional layers, ReLU and Leaky ReLU activations, max and average pooling, and more) trained for binary classification via minimization of exponential or logistic loss on separable data. In this setting, minimizing the loss necessarily means divergence of parameters (weights) to infinity, but the asymptotic behavior, i.e. the manner in which the parameters diverge, is of interest. The paper makes the following contributions: (i) It proves that the parameters converge in direction, which implies that prediction margins on the training examples converge if normalized. (ii) Under the additional assumption of locally Lipschitz gradients, it is proven that the gradient converges to the direction of the parameters. This implies margin maximization for the special cases of linear neural networks and a two layer network with squared ReLU activation. From a technical perspective, the analysis is based on a theory of non-smooth Kurdyka-Lojasiewicz inequalities for functions definable in an o-minimal structure. Experiments support the theory on both a model covered by the analysis ("homogeneous AlexNet"), and one that is not (DenseNet).

Strengths: The question of implicit regularization is of prime importance to the theory of deep learning, and the setting analyzed by this paper is very timely. In that regard, I believe the work is of high relevance to the NeurIPS community. The theoretical tools employed are very interesting, and although I did not verify all details, I believe the paper is solid from a technical perspective. I also found the text to be relatively well written, which is non-trivial given its level of mathematical depth.

Weaknesses: I have two main critiques on this work. The first relates to the significance of its results. In the setting studied, directional convergence, alignment and margin maximization have all been treated in several recent works (which the paper refers to). I know that at least in some of these works directional convergence and/or alignments were assumed (not proven), but nonetheless, my feeling is that the paper does not draw a sufficiently clear line separating itself from existing literature. For example, a very relevant existing work --- Lyu and Li 2019 --- is said to have left open the issues of directional convergence and alignment, but to my knowledge, that work does establish directional convergence, at least in some settings. I may be missing something here, but regardless, I think it is absolutely necessary to include in the current paper a detailed account for the exact differences between its results and those of existing literature. Otherwise the significance of its contributions is unclear. The second major comment I have relates to presentation. Some parts of the text (mostly in Sections 3 and 4) are extremely technical and dense, and it seems to me like much of the technicality arises from the need to account for non-smooth models (e.g. ones including ReLU activation). I think this aspect of the analysis is important, but recommend to the authors to consider deferring it to an appendix, and treating in the body of the paper only models that are differentiable. Hopefully that way a lot of the technical clutter will be avoided, and only the main ideas of the analysis will remain. Despite the above critique, I believe this paper is solid and does provide meaningful contributions. Therefore, I currently rate is as marginally above acceptance threshold, and will favorably consider increasing my score if the authors provide convincing arguments in their rebuttal. === UPDATE FOLLOWING AUTHOR FEEDBACK === I have thoroughly read the authors' feedback, as well as the other reviews. The authors have largely addressed my concerns (unclear distinction from prior work and overly technical presentation). Assuming the changes they have committed to will be incorporated, I recommend acceptance of this work.

Correctness: I did not verify all details in the analysis but I believe it is correct.

Clarity: Given the technical content I think the text is well written, but as described in "weaknesses" section above, I suggest deferring much of the technical content to an appendix. This will make the text much more readable in my opinion.

Relation to Prior Work: In my opinion relation to prior work is lacking --- see "weaknesses" section above.

Reproducibility: Yes

Additional Feedback: Minor comments: * In the CIFAR experiments, it should be described how the dataset is adapted for binary classification. * Typo in line 74: the word "section" appears twice. * Line 160: if I understand correctly, the words "to ensure it" would be more appropriate than "to rule it out". * In Lemma 3.4, projections of subdifferential are used before being defined. * Typo in line 248: "an assumptions" ==> "assumptions".


Review 2

Summary and Contributions: For binary classification with homogeneous networks and the exponential loss or logistic loss, this paper proves that the weights along gradient flow path convergence in direction. Moreover, under local Lipschitz condition on gradients, the angle between the weights and the gradients along gradient flow path is shown to approach zero. The paper also discuses the consequence of this results for implicit bias for margin maximization.

Strengths: The implicit bias of gradient decent is an important topic in the NeurIPS community. The logic of the results in many (but not all) previous papers was, for example, if it converges with a certain condition, then it satisfies KKT condition of margin maximization. While one might argue that the directional convergence part is relatively easier, it is still important to prove this precisely (and that is one of the points of mathematical proofs) and in this sense, the contribution of this paper is significant.

Weaknesses: It is not directly applicable to practical scenarios yet, partly because the paper considers gradient flow instead of finite time gradient dynamics. Also, many networks being used today are not homogeneous. However, these are understandable limitation for theoretical developments.

Correctness: All the claims seem to be correct.

Clarity: The paper is well written.

Relation to Prior Work: The paper clearly discusses how this work differs from previous contributions.

Reproducibility: Yes

Additional Feedback: I read the author response, and agree with the authors for the responses for my part. I also read other reviews and agree with the concerns for the relationship to prior work. I updated my review accordingly. Except for in section 4.2, the paper does not seem to rely on the structure of deep networks. So, main results hold for general functions that satisfy those assumptions. Also, not only the setting, but also the statement about convergence is of usual interest in many math fields. This is good and interesting for wider applicability of the results. But, it also raises a potential concern that there are related previous results in other math fields outside of machine learning literature. Sub-differential and continuous dynamics are a big field. I am not clarified to judge the novelty of the results in a wider community in this sense. If someone in the related math fields outside of machine learning literature can check its novelty, that would be nice.


Review 3

Summary and Contributions: The paper considers positively homogeneous networks and logistic or exponential losses, and assume a technical condition that the network is definable in some o-minimal structure. It studies the late training setting after the loss is smaller than 1/#datapoints. It shows that 1) weights learned by gradient flow converge in direction (implying the convergence of predictions, training errors, and the margin distribution etc); 2) if the network further has locally Lipschitz gradients, the gradients converge in direction and asymptotically align with the gradient flow path (implying margin maximization in a few settings). The analysis is by unbounded nonsmooth Kurdyka-Łojasiewicz inequalities.

Strengths: + The results are quite general and clean. It holds for a rich family of networks (ie linear, convolution, ReLU, and max-pooling layers). + The analysis technique (unbounded nonsmooth Kurdyka-Łojasiewicz inequalities) seems an interesting contribution to analyzing deep learning and nonlinear systems more broadly. (Although I'm not exactly sure its connection to the techniques in the previous work Lyu and Li [2019]).

Weaknesses: - The results are not applicable to some other important family of networks like ResNets. The analysis seems to be relying on the homogeneity. Does this mean that ResNets have different behavior? Or they have similar behaviors, but the analysis is limited by the current techniques available? - Can one have discrete-time guarantees? Especially since the analysis if for late training, one would like to have an analysis saying after how many steps the training roughly converges (stabilizes). Does it take exponential time to stabilize? - The generalization behavior is only briefly mentioned. The implications on margins are related, but more explicit discussion will be appreciated. I would also like to see more discussion on the effect of late training on generalization, e.g., does the late training after loss 1/#datapoints improve the generalization or can hurt the generalization? ============after response=============== The response clarified my questions on generalization and distinction from prior work. It doesn't adequately address the questions about discrete-time analysis and non-homogeneous networks, but I understand these two are quite beyond the scope of the paper, so I increase the score from 6 to 7.

Correctness: I believe so, though I may have only checked the key lemmas.

Clarity: Yes.

Relation to Prior Work: Yes. More discussions on generalization will be appreciated.

Reproducibility: Yes

Additional Feedback:


Review 4

Summary and Contributions: This work established that the normalized parameter vectors converge, and that under an additional assumption of locally Lipschitz gradients, the gradients also converge and align with the parameters.

Strengths: (1) This work established that the normalized parameter vectors converge; (2) Under an additional assumption of locally Lipschitz gradients, the gradients also converge and align with the parameters.

Weaknesses: (1) In terms of the results, the paper obtains a significant convergence result in the deep learning field, this work studies the binary classification problem, where the loss is the logistic loss or exponential loss. The main technical analysis is based on this. Do you think whether it is easy to consider a general loss function with some mild assumptions? (2) In terms of the assumptions, both the locally Lipchitz gradient assumption and the initial-error assumption seem strong to me. In particular, this work requires an initial risk smaller than 1/n, where n is the number of data samples. In general, the number of data samples n is typically large, which means this work requires a sufficiently good initialization. First, the initialization assumption seems a bit stronger compared with the random initialization or around-zero initialization in the literature. Second, this work also assumes perfect classification accuracy on the data samples. Despite the rationality of the assumption in practice, using this assumption along with the sufficient good initialization, it seems not hard to show the initialization is close to an optimal classifier. Could we relax such assumptions by using additional network structures such as over-parameterization? Because the convergence literature doesn't require such assumptions when assuming the over-parameterization assumption. (3) In terms of the technical analysis, this work heavily relies on previous works [Ji and Telgarsky 2018a] and [Kurdyka et al. 2000a, 2006].

Correctness: Correct.

Clarity: Yes.

Relation to Prior Work: Yes.

Reproducibility: Yes

Additional Feedback: This work studies the binary classification problem, where the loss is the logistic loss or exponential loss. The main technical analysis is based on this. Do you think whether it is easy to consider a general loss function with some mild assumptions? ==========================after response=================== I read the author response. I updated my review accordingly.

[Author Response · NeurIPS 2020]

We thank the reviewers for their comments and time.

**Reviewer #1.**   *(Relationship to prior work, particularly (Lyu and Li, 2019).)* We will expand our Related Work section
to explicitly separate our technical contribution from prior work, expanding comments found throughout this response.
As the reviewer mentions, prior work often *assumes* directional convergence and alignment, but neither indicates a
possible proof, nor even provides conclusive evidence. Regarding (Lyu and Li, 2019), they sidestepped directional
convergence by using subsequences (cf. their Theorem 4.4), and in fact they provided a pathological example where
directional convergence *fails* (cf. their Theorem J.1). Our key technical tool for directional convergence, the notion of
o-minimal definability, was only used in (Lyu and Li, 2019) to ensure a nonsmooth chain rule. Alignment, in our exact
form, does not appear in prior work, which excites us as it implies both existing and new margin maximization results.

*(Overly technical presentation.)* We agree, with hindsight, that our presentation was encumbered with too much focus
on technical details, such as how to handle nonsmooth models as mentioned by the reviewer. We will expand intuition
and machine learning connections, and move material to the appendices, as the reviewer suggests.

*(Minor comments.)* Thank you! We will address these comments in our revisions.

**Reviewer #2.**   *(Discrete-time analysis.)* We agree that a discrete-time analysis is essential. We touched upon this
in our "Concluding Remarks", but will expand the material; for instance, one can easily adapt our analysis to handle
extremely small step sizes, but handling a practical choice is much more challenging. Getting convergence rates is
another important open problem, and might be hard even for the gradient flow, despite our present work.

*(Non-homogeneous networks.)* We agree this is important, and provide illustrative support, in the form of Figure 2b on
page 3 (DenseNet) and Figure 3 in the appendix (ResNet), which does seem to suggest directional convergence holds.

*("One might argue that the directional convergence part is relatively easier".)* We will highlight in our revised Related
Work section that this question is tricky, and has stymied many mathematicians. As discussed at the end of Section 1.1
in our submission, to prove the related gradient conjecture of René Thom, mathematicians had to develop the whole
area of o-minimal structures. Even with this powerful technique, as far as we know the existing results on directional
convergence all roughly assume piecewise polynomials or real-analytic functions, and therefore cannot analyze a
nonsmooth function composed with the exp/logistic loss, which is more relevant to the deep learning community.

*("Sub-differential and continuous dynamics are a big field".)* We surveyed and cited recent work, e.g., (Davis et al.,
2020); we had to prove many new results (e.g., the unbounded nonsmooth Kurdyka-Łojasiewicz inequalities) to show
directional convergence and alignment.

**Reviewer #3.**   *(Generalization.)* We agree that generalization is essential, and will include expanded discussion in our
revisions. Briefly, on the empirical side, we point the reviewer to the large-scale experiment we cited, by Shallue et
al. (2018), which finds that even uncommonly large amounts of training do not seem to hurt generalization. On the
theoretical side, we will mention various margin-based generalization bounds, and tie them to our alignment property.

*(Relationship to (Lyu and Li, 2019).)* Our analysis of directional convergence is not relevant to this prior work: they did
not prove directional convergence but instead must use subsequences. Please refer to lines 2-9 above for more details.

*(Non-homogeneity and discrete time.)* We agree these are important; please refer to lines 14-19 above for more details.

**Reviewer #5.**   *(Regarding "(1)".)* Our analysis can be extended to many other decreasing losses with an exp tail, but
we chose to focus on the exp/logistic loss to highlight the key ideas, and moreover because the logistic loss is one of the
most widely-used losses in machine learning. Since our core technical work is on solutions at infinity, losses like the
squared loss, which imply a (finite) minimizer, require a different analysis, though some of our lemmas will still apply.

*(Regarding "(2)".)* This initialization assumption was first introduced in prior work (Lyu and Li, 2019), and allows us
to focus on the late-training regime; we can handle random initialization by first invoking a standard overparameterized
analysis as a lemma (these results only hold near random initialization). Regarding "it seems not hard to show
the initialization is close to an optimal classifier", firstly we stress this is an orthogonal concern to our directional
convergence result, which ensures gradient flow eventually stabilizes. Secondly, "optimal" in our setting is typically
"maximum margin", where it seems the late-training phase is essential, and our results handle a few such cases.

*(Regarding "(3)".)* The prior work (Ji and Telgarsky, 2018a) only considered deep linear networks, and the proofs
there share almost no techniques with the proofs of our main results, in Theorems 3.1 and 4.1. If this was a typo and
the reviewer intended (Ji and Telgarsky, 2019), then our potential function $\mathcal{J}$ in Section 4 is indeed based on their
dual potential, but their setting is linear and convex, and our nonlinear nonconvex analysis is significantly different.
Regarding the relationship to math literature, such as (Kurdyka 2000a, 2006), we had to overcome many obstacles
missing from these prior works, such as how to handle the exp/logistic loss with nonsmooth models.

[Meta-Review · NeurIPS 2020]

The reviewers agree that this paper is solving an important question with an interesting mathematical approach. The contributions, although technical, help to give rigorous justification to many works studying the behavior of neural networks or other models under gradient descent. I thus recommend accept. Do not forget to update the paper as mentioned in the rebuttal.